# Homozygous *EPRS1* missense variant causing hypomyelinating leukodystrophy-15 alters variant-distal mRNA m⁶A site accessibility

Debjit Khan [1], Iyappan Ramachandiran[1], Kommireddy Vasu[1], Arnab China[1], Krishnendu Khan[1], Fabio Cumbo [2], Dalia Halawani [1], Fulvia Terenzi[1], Isaac Zin[1,3], Briana Long[1], Gregory Costain[4], Susan Blaser[5], Amanda Carnevale[4], Valentin Gogonea[3], Ranjan Dutta [6], Daniel Blankenberg [2], Grace Yoon [4,7] ✉ & Paul L. Fox [1] ✉

Hypomyelinating leukodystrophy (HLD) is an autosomal recessive disorder characterized by defective central nervous system myelination. Exome sequencing of two siblings with severe cognitive and motor impairment and progressive hypomyelination characteristic of HLD revealed homozygosity for a missense single-nucleotide variant (SNV) in *EPRS1* (c.4444 C > A; p.Pro1482Thr), encoding glutamyl-prolyl-tRNA synthetase, consistent with HLD15. Patient lymphoblastoid cell lines express markedly reduced EPRS1 protein due to dual defects in nuclear export and cytoplasmic translation of variant *EPRS1* mRNA. Variant mRNA exhibits reduced METTL3 methyltransferase-mediated writing of *N*⁶-methyladenosine (m⁶A) and reduced reading by YTHDC1 and YTHDF1/3 required for efficient mRNA nuclear export and translation, respectively. In contrast to current models, the variant does not alter the sequence of m⁶A target sites, but instead reduces their accessibility for modification. The defect was rescued by antisense morpholinos predicted to expose m⁶A sites on target *EPRS1* mRNA, or by m⁶A modification of the mRNA by METTL3-dCas13b, a targeted RNA methylation editor. Our bioinformatic analysis predicts widespread occurrence of SNVs associated with human health and disease that similarly alter accessibility of distal mRNA m⁶A sites. These results reveal a new RNA-dependent etiologic mechanism by which SNVs can influence gene expression and disease, consequently generating opportunities for personalized, RNA-based therapeutics targeting these disorders.

The essential function of the 20 cytosolic aminoacyl-tRNA synthetases (aaRSs) is high-fidelity decoding of genetic information carried by mRNA during protein synthesis. The aaRSs catalyze ATP-dependent charging of tRNAs with cognate amino acids for delivery to the ribosome A-site. Pathogenic variants in cytosolic aaRSs are associated with several neurological disorders with myelination defects including epileptic encephalopathy, progressive microcephaly, Charcot-Marie-Tooth disease, and hypomyelinating leukodystrophy (HLD)[1], and there is emerging interest in aaRSs as potential therapeutics and therapeutic targets[2]. Nine of the twenty aaRS activities [in eight proteins since EPRS1 contains two covalently linked synthetase domains, glutamyl-tRNA synthetase (GluRS) and prolyl-tRNA synthetase (ProRS)] reside in

a cytoplasmic multi-tRNA synthetase complex (MSC) with three non-synthetase proteins, AIMP-1, −2, and −3[3,4]. Remarkably, pathogenic variants in seven of eleven MSC constituents cause a broad spectrum of neurological diseases. Pathogenic variants in *QARS1* and *KARS1* cause progressive microcephaly[5,6], peripheral neuropathy[7], and progressive leukoencephalopathy with brainstem and spinal cord calcifications[8]. Additionally, bi-allelic missense variants in genes encoding three other aaRSs−*RARS1*[9], *DARS1*[10], and *EPRS1*[11], cause childhood-onset HLD, specifically HLD9, HBSL (hypomyelination with brain stem and spinal cord involvement and leg spasticity), and HLD15, respectively. Variants in two non-aaRS MSC components−*AIMP1*[12] and *AIMP2*[13]−also cause white matter disorders, but the underlying pathology is demyelination secondary to neurodegeneration.

The leukodystrophies are a family of more than 50 distinct heritable central nervous system (CNS) disorders characterized by diminished cerebral and cerebellar white matter due to dysregulated myelin formation or degeneration[14]. HLD is rare, but comprises the single largest category among undiagnosed genetic leukodystrophies, which collectively impacts ~1 in 7500 live births, representing a major group of neurodevelopmental disorders[15]. Clinical features include severe cognitive and motor impairment with onset in early childhood or adolescence. At present there are no curative treatments; patient management includes serial brain MRI to monitor hypomyelination, genetic testing to elucidate etiology, and symptomatic treatment of neurologic and other medical complications[16]. The causative role of multiple aaRS variants in HLD is well-established, however, a unified hypothesis delineating the mechanism by which variant aaRSs cause HLD has not yet emerged. For nearly all *EPRS1* and *RARS1* variants, the relevant aminoacyl charging activity in patient fibroblasts is reduced by about 30–50% compared to healthy controls[11,17]. In one study, the reduced charging activity in fibroblasts from patients with *EPRS1* variants was due to reduced amount of enzyme, as well as reduced specific charging activity of recombinant protein[11]. The decrease in cellular RARS1 protein ranged from barely discernable by immunoblot to ~80% in patients with *RARS1* variants[9,17]. In most cases, decreased specific charging activities are attributable to variants in or near catalytic or tRNA-binding sites[10,11]. However, the specific mechanism underlying reduced variant aaRS expression has not yet been elucidated.

We have identified a homozygous c.4444 C > A; p.Pro1482Thr missense single nucleotide variant (SNV, rs930995541) in *EPRS1* in two siblings presenting with clinical features consistent with childhood-onset HLD15 (OMIM 617951). Here, we show the variant reduces EPRS1 expression in patient cells by inhibiting m6A modification of requisite mRNA target sites. Unexpectedly, the variant does not alter m6A site sequence, as observed for other disease-associated genetic variants, but instead masks accessibility of variant-distal mRNA m6A sites. Importantly, bioinformatic analysis suggests widespread disease-associated SNVs that also influence distal m6A site accessibility, and a novel etiologic principle of genetic disease with potential for personalized, RNA-based therapeutics.

## Results

### Evaluation of siblings with global developmental delay and neurological impairment

Two siblings born to healthy consanguineous parents of Pakistani descent were evaluated in the Neurogenetics Clinic at the Hospital for Sick Children (Fig. 1a). The Proband, Sibling 1, is an 18-year-old male with severe global developmental delays and intellectual disability, ataxia, microcephaly, rotatory nystagmus, axial hypotonia, and progressive bilateral lower limb spasticity on serial neurological examinations. At 18 years, he was fully dependent on a wheelchair for ambulation, and his cognitive function was estimated to be at the level of a 2-year-old. Serial magnetic resonance imaging (MRI) of the brain at 2 and 9 years showed microcephaly with diffuse supratentorial and infratentorial volume loss, thinning of the corpus callosum, and global

hypomyelination with progressive myelin loss (Fig. 1b). The proband's sister, Sibling 2, exhibited similar clinical features. At 16 years, her primary mode of ambulation was a wheelchair, but she could use a walker on level ground. She was less severely affected cognitively, but like her brother was completely dependent for all activities of daily living. At age 15 she became non-ambulatory and was also diagnosed with premature ovarian insufficiency for which she is treated with levonorgestrel-ethinyl estradiol. MRI of the brain at 3 and 8 years showed microcephaly and global hypomyelination with progressive myelin loss similar to Sibling 1 (Fig. 1b). Further clinical details are available in Supplementary Tables 1, 2. Clinical, quad-based exome sequencing of the proband and Sibling 2 revealed homozygosity for a novel missense variant in the gene *EPRS1* [NM_004446.2: c.4444 C > A; p.(Pro1482Thr)]. Both parents and their unaffected son are heterozygous for this variant, and no alternative diagnosis was identified by whole-exome sequencing. Immortalized lymphoblastoid cell lines (LCLs) were generated from affected siblings, carrier parents, and unrelated controls, transformed with Epstein-Barr virus, and the *EPRS1* variant was validated by Sanger sequencing[18] (Fig. 1c).

### Dual post-transcriptional mechanisms drive low EPRS1[P1482T] expression

The Pro1482Thr substitution is in the $Zn^{2+}$-binding domain of EPRS1, distant in sequence space from the catalytic and anti-codon binding domains[19,20] (Fig. 2a, top), but spatially near the intersection of the domains according to the X-ray structure of the human ProRS dimer (Fig. 2a, bottom)[19]. Pro[1482] is in a highly conserved region and is present in all species investigated including *S. cerevisiae*, and possibly *T. thermophilus* (Supplementary Fig. 1a). Because EPRS1 is a unique bifunctional synthetase with covalently linked ProRS and GluRS activities, both activities were determined in LCL lysates by charging yeast tRNA with [14C]Pro or [14C]Glu[21]. Cellular ProRS and GluRS charging activities by LCLs from affected siblings were about 20% of unaffected controls; carrier parent LCLs exhibited intermediate activities (Fig. 2b, left and middle panels). tRNA charging of [14C]Phe by FARS1 was comparable for all LCLs, indicating specificity of the inhibition of ProRS and GluRS charging (Fig. 2b, right panel). Reduced cellular ProRS and GluRS activities might reflect either decreased amount of EPRS1[Pro1482Thr] or reduced specific charging activity. To test the latter mechanism, recombinant N-terminal, FLAG-tagged wild-type (WT) and Pro1482Thr mutant EPRS1 were expressed and purified from HEK293F cells. Specific ProRS and GluRS charging activities were determined in vitro as incorporation of [14C]Pro and [14C]Glu, respectively, into yeast tRNA[21]. Specific activities of both catalytic domains were identical in WT and mutant EPRS1 (Fig. 2c). EPRS1 forms functional dimers via ProRS domain interactions[19]. Molecular dynamic simulation (MDS), comparing wild-type and Pro1482Thr mutant ProRS dimers, revealed only minor structural differences in the catalytic and anticodon-binding domains, but somewhat larger differences in the zinc-binding domains in monomer A (Supplementary Fig. 2a, b). Both simulated zinc-binding domains were shifted compared to the crystal structure. Pro1482 terminates a β-sheet, and Thr substitution does not perceptibly alter the conformation of this secondary structure (Supplementary Fig. 2c). Likewise, localization of key elements within the monomer B structure, i.e., ATP, $Mg^{2+}$ ions, proline substrate, and $Zn^{2+}$ ions are not markedly altered (Supplementary Fig. 2d). These results suggest that diminished ProRS charging activity observed in sibling LCLs is not due to reduced specific activity but implicates differences in EPRS1 level.

EPRS1 amount in LCL lysates was determined by immunoblot using antibody targeting the linker region. EPRS1 levels in the siblings was ~20% of that in unrelated controls bearing WT *EPRS1*; the carrier parents exhibited ~50–60% of control levels (Fig. 2d). Importantly, *EPRS1* mRNA expression was not significantly different in control, parent, and sibling LCLs (Fig. 2e). Similarly, 3′-RACE analysis of *EPRS1* mRNA in LCLs, from the next-to-last exon to the poly-A tail, and

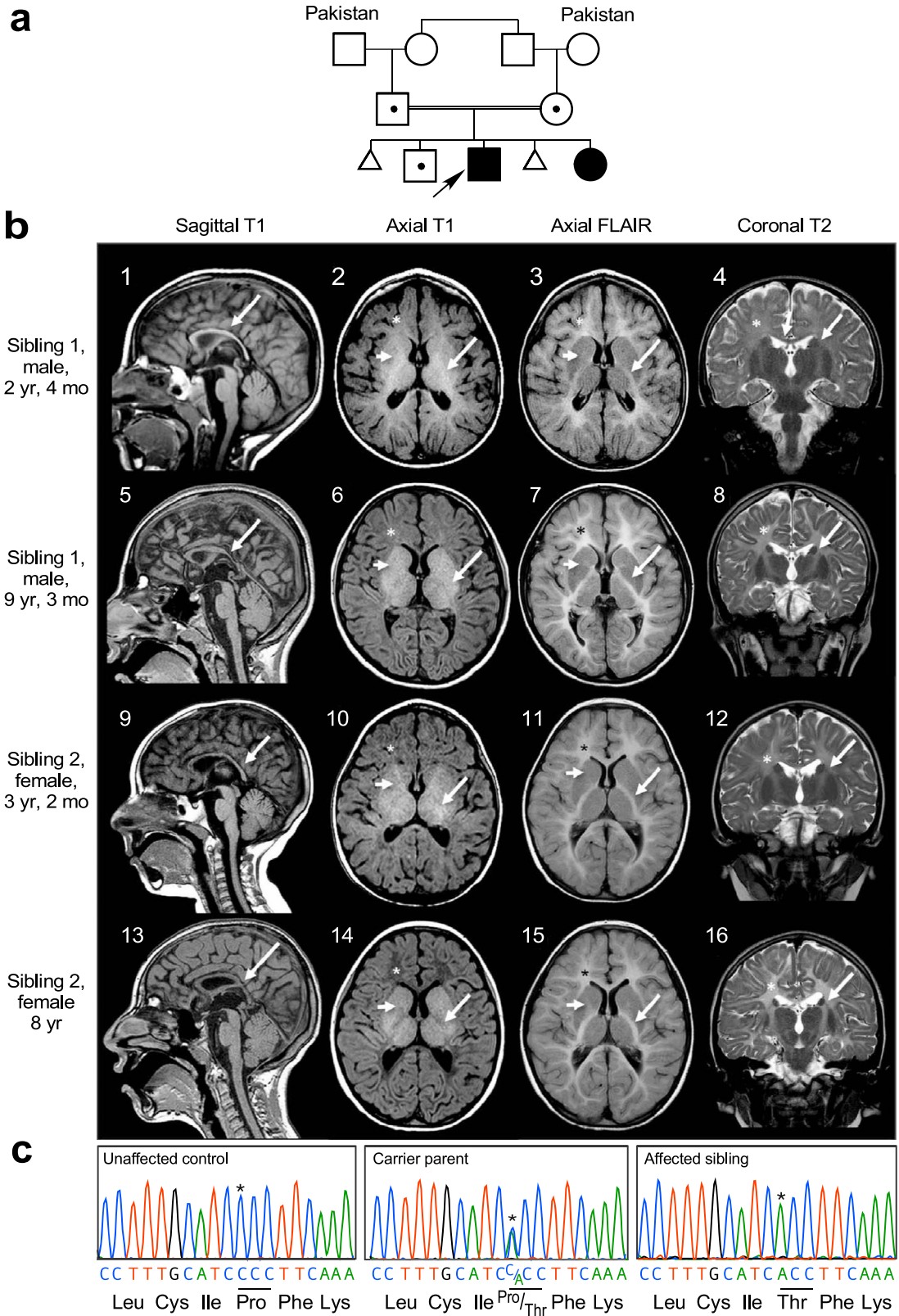

spanning the c.4444 C > A variant in the terminal exon, revealed identical splicing and polyadenylation (Supplementary Fig. 1b, c). These results indicate post-transcriptional regulation is responsible for reduced expression of sibling EPRS1. In view of the essential role of aaRSs in protein synthesis, the effect of the ~80% decrease in EPRS1 amount on global protein synthesis in LCLs from the siblings was determined by polysome profiling. Polysome profiles comparing protein synthesis in LCLs from the female sibling to the mother (Fig. 2f, left) and male sibling to the father (Fig. 2f, right) are virtually identical. Likewise, metabolic labeling with [$^{35}$S]Cys/Met confirmed that global protein synthesis, as shown by labeling nascent protein, is not reduced in sibling LCLs indicating an ~80% inhibition of EPRS1 expression is not injurious to cells (Fig. 2g), and consistent with undiminished specific activity of the EPRS$^{P1482T}$ variant.

**Fig. 1 | Clinical and genetic characterization of siblings with HLD. a** Family pedigree; arrow indicates proband. **b** Top row (Sibling 1 at 2 yr 4 mo): Sagittal T1 image (1) shows a thin, myelin-mature (arrow) corpus callosum (CC). Axial T1 (2) reveals myelin signal in the anterior limb of the internal capsule (ALIC, short arrow) and in the posterior limb of the internal capsule (PLIC, arrow). Faint myelin signal is present in frontal (*) and parietal lobes on T1 axial images (2). Axial FLAIR (3) and coronal T2 (4) images show abnormally increased signal (arrows) in the PLIC, frontal lobe white matter (*), and parietal white matter. Second row (Sibling 1 at 9 yr 3 mo): Sagittal T1 (5) reveals thin CC with myelin signal loss (arrow). Frontal lobe white matter (*) has lost signal on T1 axial (6), and is unchanged on axial FLAIR (7) and coronal T2 (8) images. ALIC (short arrows) has thinned on T1 (6), FLAIR (7), and T2 (8) weighted axial images, while PLIC has thinned on T1 (6) and remains abnormal in signal on T2 (7) and FLAIR (8). Both the ALIC and genu of internal capsule (short arrows) have lost myelin signal on T1 (6), FLAIR (7), and T2 (8) weighted images. Third row (Sibling 2 at 3 yr 2 mo): Sagittal T1 image (9) shows a thin CC with trace myelin in splenium (arrow). Axial T1 (10) also reveals trace myelin signal (arrows) in the ALIC and in the corticospinal tract in the mid-third of the PLIC. Myelin signal is absent in the frontal (*) and parietal lobes on T1 axial images (10). Axial FLAIR (11) and coronal T2 (12) images show abnormally increased signal (arrows) in the PLIC (arrows), frontal lobe white matter (*), and parietal white matter. Bottom row (Sibling 2 at 8 yr): Sagittal T1 (13) reveals persistently thin CC with loss of splenium myelin (arrow). Frontal lobe white matter (*) is lower in signal on T1 axial (14), and unchanged on axial FLAIR (15) and coronal T2 (16) images. PLIC signal abnormality is unchanged. ALIC has lost myelin signal on T1 (14), FLAIR (15), and T2 (16) weighted images. **c** Sanger sequencing of DNA isolated from immortalized LCLs generated from unaffected control 3 (left), carrier parent (center), and affected sibling (right). EPRS1 c4444C>A variant position (*).

The fate and function of EPRS1 protein could be influenced by its dimerization status, as well as by localization outside the MSC[19,22–24]. Size fractionation of recombinant WT and Pro1482Thr ProRS showed similar extents of dimerization (Fig. 2h). To determine the influence of the EPRS1[P1482T] substitution on residence in the MSC, FLAG-tagged, full-length WT and mutant *EPRS1* cDNAs were transfected into HEK293T cells. EPRS1 was isolated from lysates with anti-FLAG resin, eluted with FLAG peptide, and subjected to immunoblot. WT and EPRS1[P1482T] bind equally to the MSC constituents tested, indicating normal MSC incorporation of the mutant (Fig. 2i). The stability of mutant EPRS1 was investigated directly. LCLs were treated with cycloheximide to block protein synthesis, and EPRS1 disappearance monitored by immunoblot. No loss was detected over a 24-h period in any LCL, indicating all EPRS1 forms are highly stable (Fig. 2j). The translation state of WT and variant *EPRS1* mRNA was explored by polysome profiling. An ~40% reduction in polysomal *EPRS1* mRNA was observed in affected sibling LCLs, consistent with an important contribution of translation to reduced EPRS1 expression (Fig. 2k); however, the amount of reduction is less than that of the steady-state level of protein, suggesting additional mechanisms might be operative. A possible defect in nuclear export of newly transcribed c.4444 C > A *EPRS1* mRNA was explored by fractionation of LCL lysates into nuclear and cytoplasmic pools. The ratio of nuclear to cytoplasmic *EPRS1* mRNA, as measured by primer/probe spanning the exon 3–4 junction, was ~2-fold higher in siblings compared to controls, and the parental level was intermediate, consistent with higher nuclear levels of c.4444 C > A *EPRS1* mRNA and diminished export (Fig. 2l). Together, these results indicate EPRS1 expression in sibling cells is reduced by dual post-transcriptional mechanisms.

## Role of *EPRS1* mRNA m⁶A modification in reduced expression of variant EPRS1

To investigate the mechanism underlying low expression of variant EPRS1, chimeric reporters were generated containing hRLuc upstream of the 3′-terminal region of EPRS1 mRNA surrounding the variant site, namely, exons 31 and 32, bearing either WT or c.4444 C > A sites (hRLuc-EE) (Fig. 3a, left-top). In all reporters, the EPRS1 RNA sequence was in-frame with hRLuc, and without an intervening stop codon. Following transfection into HEK293T cells, expression of the hRLuc-EE reporter bearing the c.4444 C > A variant was ~25% less than the WT—a lower level of inhibition than observed for endogenous *EPRS1* mRNA (Fig. 3a, right). Because splicing facilitates nuclear mRNA export in mammalian cells[25], we generated a reporter pair containing the intervening intron, I31 (hRLuc-EIE) (Fig. 3a, left-middle). Inclusion of the intron induced a 50% decrease in expression of the reporter bearing the c.4444 C > A variant. A similar result was observed following transfection into control LCLs, indicative of a cell type-independent effect of the variant (Supplementary Fig. 3a). Introduction of a second upstream intron between exons 30 and 31 (hRLuc-E-RBGI-EIE, rabbit β-globin gene intron was used because the ~3 kb *EPRS1* intron 30

contains a potential insertion sequence/transposon element with multiple inverted repeats) did not further reduce relative expression of the reporter bearing the c.4444 C > A variant (Fig. 3a). To determine if sequences within the intervening intron influence expression, I31 was replaced by an unrelated chimeric intron, cI (Fig. 3b, Supplementary Fig. 3b)[26]. Although expression of the non-mutated reporter was reduced, expression of hRLuc-EcIE and hRLuc-E-RBGI-EcIE reporters bearing the c.4444 C > A variant was inhibited to the same extent as reporters containing I31, i.e., hRLuc-EIE. An intron (e.g., cI) in the 5′UTR of hRLuc-EE reporter (Supplementary Fig. 3c, left, top two schematics), does not synergize with the HLD-causing c.4444 C > A variant (Supplementary Fig. 3c, right). Synergy is evident only when an intervening intron (e.g., I31) is present between exons 31 and 32 (Supplementary Fig. 3c, left, bottom two schematics), highlighting the specific role of this exon-exon junction. Possibly, the intron guides specificity of m⁶A methylation at physiologically relevant sites in the gene-end architecture, spatially regulated by deposition of exon-junction complexes[27]. Interplay of nuclear reader YTHDC1 with splice adapters and the mRNA export pathway has been reported[28]. hRLuc-EIE was selected for subsequent reporter-based experiments.

The c.4444 C > A variant site is in the terminal exon, exon 32, 55 nucleotides downstream of the junction with exon 31 (Fig. 3c). Importantly, with respect to mRNA architecture, this is a hotspot for methylation of the $N^6$-position of adenosine (m⁶A, $N^6$-methyladenosine), the most abundant internal mRNA modification[29,30]. Global analysis revealed more than 70% of all m⁶A residues in mRNAs are in the 3′-most exon, peaking just downstream of the exon start[31]. Likewise, sequences recognized for METTL3-dependent m⁶A modification, i.e., *DRACH* (A/G/U-A/G-A-C-A/C/U) sequences, are enriched in terminal exons[31,32]. m⁶A modification and its cellular consequences are dictated by sequence-specific writers, erasers, and readers[33]. Importantly, occupancy of m⁶A-modified sites determines both nuclear export and translation, as well as mRNA stability[27,28,32,34–36]. Human *EPRS1* mRNA exhibited three experimentally confirmed m⁶A sites in the region near the c.4444 C > A site in m⁶A-Atlas (version 1): an upstream site in exon 31 (16728), and two downstream sites in terminal exon 32—one in the coding region (16727), and another in the 3′-UTR (16726) (Fig. 3c)[37]. The potential role of m⁶A in determining EPRS1 expression was investigated by knockdown of METTL3, the catalytic component of the principal m⁶A writer complex[34]. METTL3 knockdown in HEK293T cells markedly reduced EPRS1 expression (Fig. 3d, Supplementary Fig. 4a). YTHDC1 is a member of a family YTH domain-containing proteins that are m⁶A readers that regulate mRNA stability and translation, as well as nuclear export[33]. Specifically, YTHDC1 is a nuclear reader of m⁶A-modified mRNA that regulates nuclear mRNA export[34]. siRNA-mediated knockdown of YTHDC1 in 293T cells (Supplementary Fig. 4b) and control LCLs (Fig. 3e, left and Supplementary Fig. 4c) inhibited EPRS1 expression, implicating m⁶A modification in export of *EPRS1* mRNA. In control LCLs, YTHDC1 knockdown exacerbated nuclear retention of *EPRS1* mRNA as shown by cell fractionation

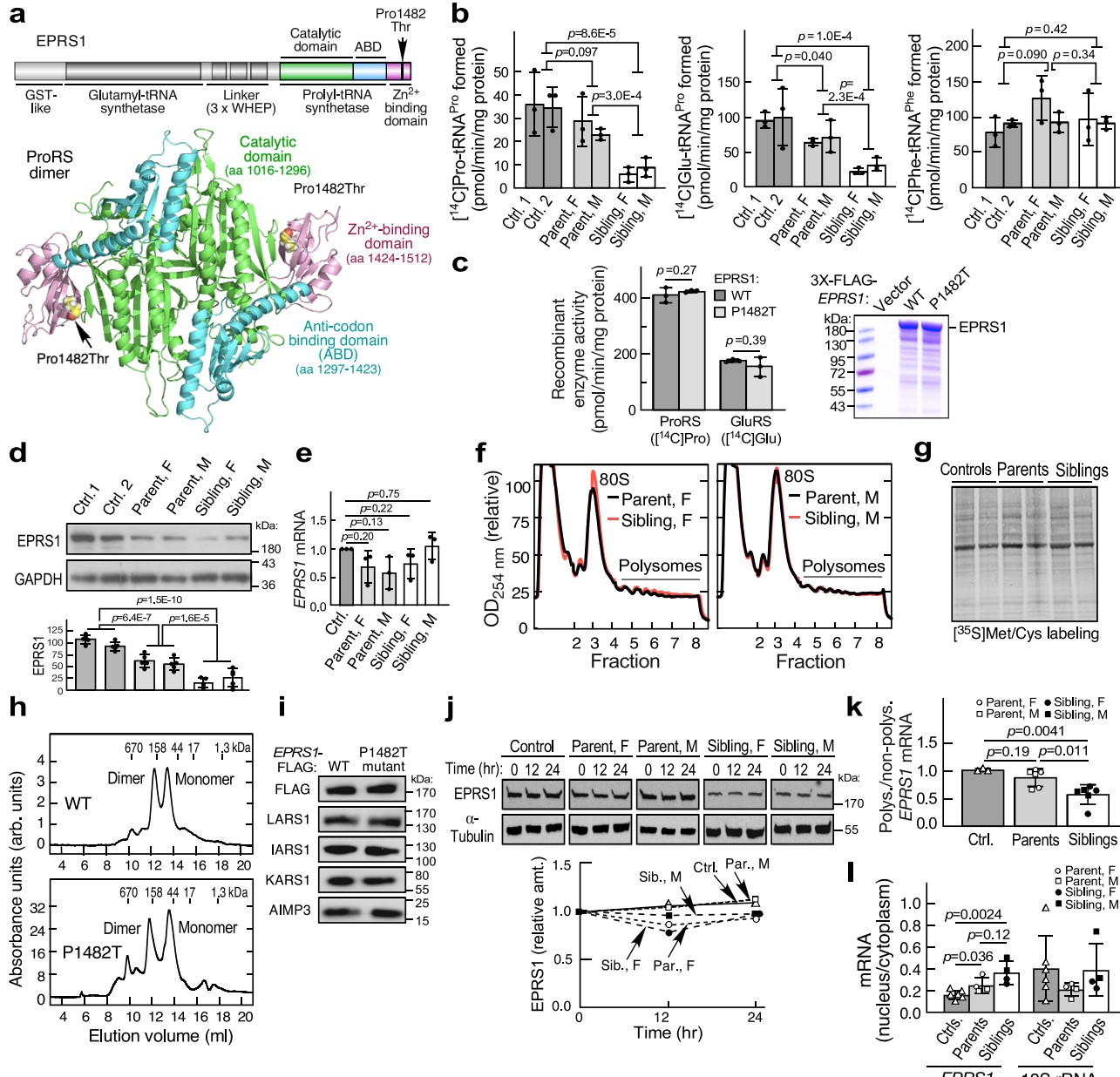

**Fig. 2 | Dual post-transcriptional mechanisms dictate reduced expression of Pro1482Thr EPRS1 in siblings with HLD. a** Domain structure of EPRS1 including ProRS sub-domains and Pro1482Thr substitution site (top). Crystal structure of ProRS dimer of human EPRS1 highlighting Pro1482Thr variant (bottom). **b** Cell tRNA charging activities of ProRS (left), GluRS (center), and FARS1 (right) were determined in LCL lysates by charging yeast tRNA with [$^{14}$C]Pro, [$^{14}$C]Glu, and [$^{14}$C]Phe, respectively. Mean ± SD, $n = 3$ biological replicates. $p$ values are from unpaired two-tailed $t$-test. **c** Recombinant WT and Pro1482Thr (P1482T) mutant human 3xFLAG-EPRS1 were expressed in HEK293F cells and detected by Imperial stain (right). ProRS and GluRS specific tRNA charging activities were determined by incorporation of [$^{14}$C]Pro and [$^{14}$C]Glu, respectively, into yeast tRNA (left). Mean ± SD, $n = 3$ biological replicates; $p$ values are from unpaired two-tailed $t$-test. **d** EPRS1 in LCL lysates was determined by immunoblot and densitometry. Mean ± SD, $n = 4$ biological replicates; $p$ values are from unpaired two-tailed $t$-test. **e** EPRS1 mRNA was determined by RT-qPCR and normalized to GAPDH mRNA. Mean ± SD, $n = 3$ biological replicates; $p$ values are from one-sample $t$-test (two-tailed), fold-change compared to control. **f** Polysome profiling comparing LCLs from female sibling and mother (left), and comparing LCLs from male sibling and father (right). **g** Metabolic

labeling of LCLs with [$^{35}$S]Cys/Met, followed by SDS-PAGE and autoradiography. **h** Dimerization status of ProRS domain. Recombinant WT (top) or Pro1482Thr (bottom) ProRS purified from *E. coli* and analyzed by size-exclusion chromatography. **i** Incorporation of mutant EPRS1 in the MSC was determined following transfection of pCMV vectors expressing FLAG-tagged, full-length WT and Pro1482Thr mutant EPRS1 into HEK293T cells. Tagged protein was purified with anti-FLAG resin, and bound proteins eluted and subjected to immunoblot. **j** Time course of EPRS1 expression in cycloheximide-treated LCLs (top). Densitometric quantitation of EPRS1 expression (bottom). **k** Determination of EPRS1 mRNA translation in LCLs by polysome profiling. Mean ± SD, $n = 3$ biological replicates for control LCL and $n = 6$ pooled female and male biological replicates for parent and sibling LCLs; $p$ values are from unpaired two-tailed $t$-test. **l** Nuclear and cytoplasmic fractions of LCL lysates probed for EPRS1 mRNA and 18S rRNA by RT-qPCR. Mean ± SD, $n = 6$ pooled biological replicates for control LCLs, $n = 4$ pooled female and male biological replicates for parent and sibling LCLs; $p$ values are from unpaired two-tailed $t$-test. Source data are provided in figshare repository [https://doi.org/10.6084/m9.figshare.25607931].

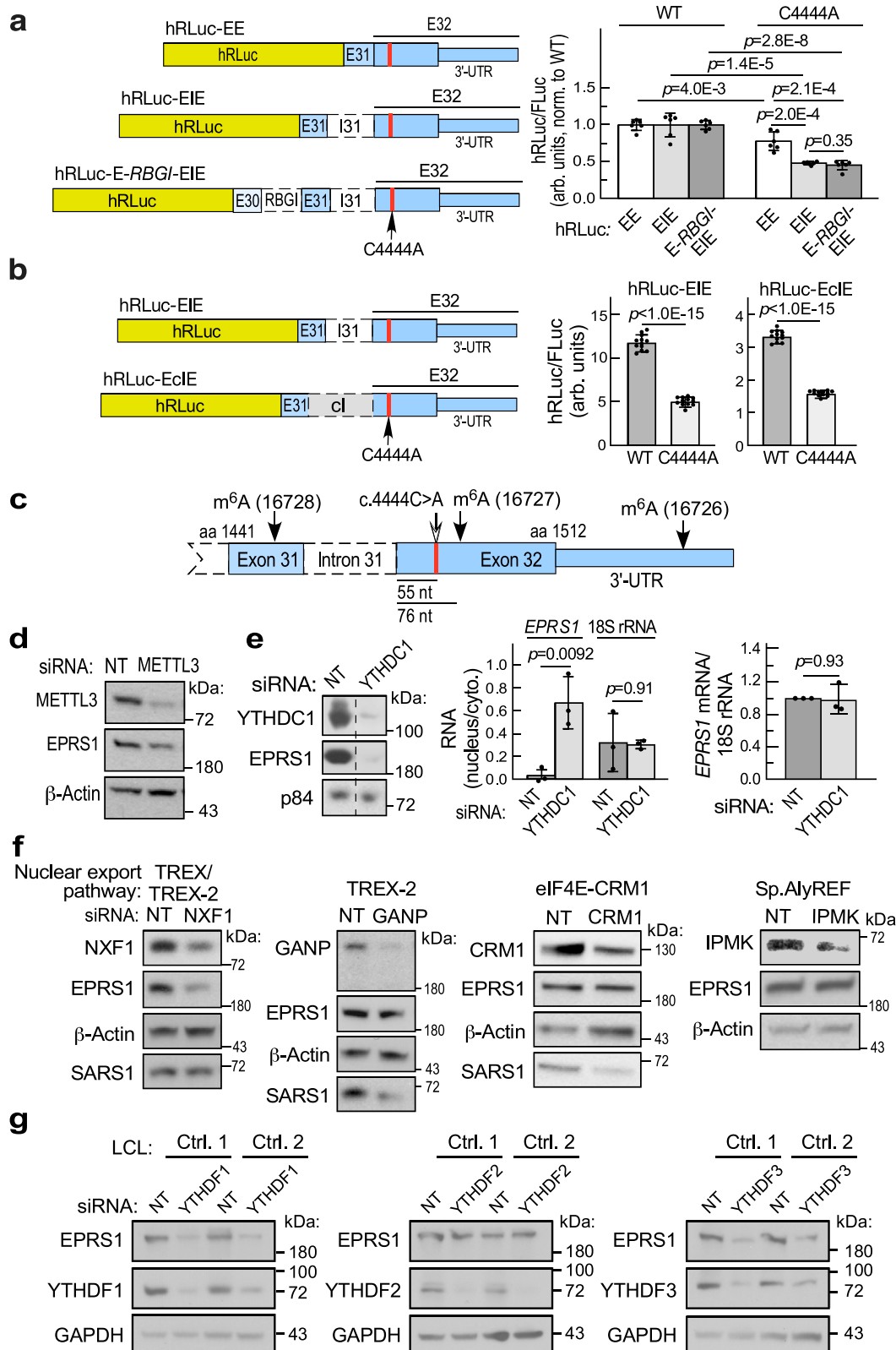

and RT-qPCR (Fig. 3e, center), but did not influence total *EPRS1* mRNA (Fig. 3e, right).

Nuclear mRNAs are packaged into messenger ribonucleoprotein complexes and exported from the nucleus via a family of nuclear pore complexes embedded in the nuclear envelope[38]. Importantly, the transcription-export complex (TREX), in association with YTHDC1, has primary responsibility for nuclear export of m[6]A-modified mRNAs[34].

To determine the nuclear RNA exporter responsible for *EPRS1* mRNA export, specific constituents of nuclear exporters were subjected to siRNA-mediated knockdown[39]. Knockdown of NXF1 (nuclear RNA export factor 1), an integral component of TREX, markedly inhibited expression of EPRS1; whereas seryl-tRNA synthetase (SARS1) expression was not inhibited (Fig. 3f, left and Supplementary Fig. 5a). Knockdown of GANP (germinal center-associated nuclear protein) of

**Fig. 3 | Role of m⁶A modification in EPRS1 mRNA nuclear export and expression. a** Role of terminal intron in EPRS1 reporter expression. hRLuc reporters containing terminal exons 31 and 32 (hRLuc-EE), or terminal exons with intervening intron, I31 (hRLuc-EIE), or hRLuc-EIE reporter with exon 30 and intervening intron, RBGI (hRLuc-E-RBGI-EIE); all reporters with or without C4444A mutation (left). Following transfection into HEK293T cells, hRLuc activities were normalized to FLuc (right). Mean ± SD, $n = 6$ biological replicates; $p$ values are from unpaired two-tailed $t$-test. **b** Effect of replacement of I31 with alternate intron, cI (hRLuc-EcIE, left). Normalized reporter activities of reporters with and without C4444A mutation (right). Mean ± SD, n = 12 biological replicates; $p$ values are from unpaired two-tailed $t$-test. **c** Experimentally validated m⁶A sites in exons 31-32 in human *EPRS1* pre-mRNA. **d** Following siRNA-mediated knockdown of METTL3 in HEK293T cells EPRS1 expression in lysates was determined by immunoblot. **e** YTHDC1 knockdown in LCLs decreased EPRS1 expression (left) and *EPRS1* mRNA nuclear retention (center), but not *EPRS1* mRNA amount (right). Dashed line indicates removal of a single lane; data are from the same gel as shown in Supplementary Fig. 4c. Mean ± SD, $n = 3$ biological replicates; $p$ values are from one-sample two-tailed $t$-test, compared to non-targeting siRNA. **f** Effect of inhibition of NXF1 in TREX/TREX-2 (left), GANP in TREX-2 (2nd from left), CRM1 in eIF4E-CRM1 (3rd from left) and IPMK in Sp.AlyREF (right) export pathways on EPRS1 expression in HEK293T cells. **g** Influence of YTHDF readers on EPRS1 expression. LCLs were subjected to siRNA-mediated knockdown targeting YTHDF1 (left), YTHDF2 (center), and YTHDF3 (right); $p$ values are from unpaired two-tailed $t$-test. For immunoblots in (**d**–**g**), refer to Supplementary Figs. 4–6 for densitometric quantification across biological replicates. Source data are provided in figshare repository [https://doi.org/10.6084/m9.figshare.25607931].

the transcription-export complex-2 pathway, CRM1 (chromosomal maintenance 1) of the eIF4E-CRM1 pathway, or IPMK (inositol polyphosphate multikinase) of a specialized AlyREF pathway, did not inhibit EPRS1 expression, but interestingly, the first two inhibited expression of SARS1 (Fig. 3f, right 3 panels and Supplementary Fig. 5b–d). The results implicate the METTL3-YTHDC1-TREX pathway as the nuclear export pathway utilized by m⁶A-modified *EPRS1* mRNA, and suggest that the c.4444 C > A variant negatively influences the function of one or more pathway constituents.

The YTHDF series of cytoplasmic m⁶A readers (YTHDF1/2/3) facilitate translation and mRNA stability. However, there is uncertainty on the relative importance of these reader functions[36,40]. Knockdown of YTHDF1 and YTHDF3 in control LCLs markedly reduced EPRS1 expression, whereas knockdown of YTHDF2 had no effect (Fig. 3g and Supplementary Fig. 6a–c). To determine relative and additive roles of nuclear and cytoplasmic m⁶A readers as determinants of EPRS1 expression, double knockdown experiments, i.e., siYTHDC1 + siYTHDF1 and siYTHDC1 + siYTHDF3, were done in control LCLs. Individual knockdowns of YTHDC1 and YTHDF1 decreased EPRS1 expression by ~70%; added in combination expression was reduced by nearly 90% (Supplementary Fig. 7a). Likewise, individual knockdown of YTHDC1 and YTHDF3 comparably reduced EPRS1 expression, but double knockdown compounded the inhibition (Supplementary Fig. 7b). Thus, optimal EPRS1 expression in healthy cells requires both YTHDC1-mediated nuclear export via NXF1, followed by YTHDF1/3-assisted translation.

## Identification of *EPRS1* mRNA m⁶A sites influenced by the c.4444 C > A variant

To facilitate investigation of m⁶A modification of the *EPRS1* reporter, background m⁶A modification of hRLuc RNA was reduced by generating constructs in which the eight *DRACH* sites in hRLuc were nullified by synonymous mutation, except for an obligate Thr184Ser mutation (Fig. 4a, left; Supplementary Fig. 8). Following transfection into HEK293T cells, activity of the DRACH⁻-containing WT construct was slightly lower than the DRACH⁺ reporter, possibly due to the non-synonymous substitution near the active site[41]; however, reduced expression by c.4444 C > A variant was retained, or possibly exacerbated (Fig. 4a, right), enabling an assay for m⁶A modification. To determine the specific m⁶A site (or sites) contributing to EPRS1 expression, the three known *EPRS1* m⁶A sites were pairwise inactivated by mutation in the DRACH⁻ hRLuc construct. Sites 16727 and 16728 were disrupted by synonymous mutations, while 3′UTR-site 16726 was disrupted by minimally altering the minimum energy-predicted RNA structure. In the context of the WT C4444 sequence, simultaneous disruption of the 16727 and 16728 sites (thus permitting modification of the 16726 site only) almost completely blocked hRLuc expression, but mutation of the other pairs, 16726/16728 and 16726/16727, did not reduce expression (Fig. 4b). Virtually identical results were observed in the U87-MG glioblastoma cell line (Supplementary Fig. 9a). This result indicates that m⁶A modification of either 16727 or 16728, the sites flanking the c.4444 C > A site, are sufficient to induce hRLuc

expression, but mutation of both sites prevents expression. Similar results in 293 T and U87-MG cells indicate the mechanism is cell type-independent. The role of the c.4444 C > A variant in m⁶A modification was directly assessed by methylated RNA-immunoprecipitation (meRIP) in which the DRACH⁻ hRLuc reporter was transfected into HEK293T cells and subjected to immunoprecipitation with anti-m⁶A antibody, followed by RT-qPCR using primers for hRLuc. Elimination of both 16727 and 16728 sites reduced m⁶A modification of the DRACH⁻ hRLuc reporter in HEK293T cells and in U87-MG cells (Fig. 4c and Supplementary Fig. 9b). Additionally, mutation of the C4444 site inhibited m⁶A modification to about the same extent as m⁶A site mutation. This experiment validates m⁶A modification of the *EPRS1* mRNA reporter and shows that the c.4444 C > A variant inhibits m⁶A modification of critical sites responsible for EPRS1 expression.

The specific m⁶A-related defect that reduces expression of variant EPRS1 was investigated in patient LCLs. The steady-state levels of m⁶A writers, readers, and erasers were unchanged in patient LCLs (Fig. 4d). In a meRIP-qPCR approach, anti-m⁶A antibody pulldown of c.4444 C > A variant *EPRS1* mRNA was reduced by ~75% compared to controls, confirming reduced m⁶A modification of endogenous *EPRS1* mRNA (Fig. 4e). Variant *EPRS1* mRNA exhibited reduced interaction with YTHDC1, the m⁶A reader that directs mRNA nuclear export, confirming the reporter experiments (Fig. 4f). YTHDC2 and YTHDF2 are nuclear and cytoplasmic m⁶A readers, respectively, that regulate mRNA stability[42,43]; in addition, YTHDF2 stimulates cap-independent translation-initiation upon heat shock stress[44] and YTHDC2 stimulates translation-elongation by resolving secondary structures in coding sequence[45]. Neither protein exhibited differential binding to variant *EPRS1* mRNA (Fig. 4g). YTHDF1 and YTHDF3 facilitate translation of bound mRNAs[46,47]. YTHDF3 tunes the translation-activating role of YTHDF1 on m⁶A-modified RNA, and can influence mRNA stability in conjunction with YTHDF2[48]. Recently, YTHDFs 1 and 3, like YTHDF2, have been implicated in mRNA degradation[36]. Importantly, binding of YTHDF1 and YTHDF3 to variant *EPRS1* mRNA was diminished compared to wild-type mRNA (Fig. 4h). Supporting our findings, variant-proximal RNA regions bind YTHDC1 and YTHDF1, in experimentally determined RNA-protein interaction datasets in CLIPdb[49]. Remarkably, a single point mutation near the stop codon of *EPRS1* mRNA reduces m⁶A modification at two sites, inhibits binding of three YTH domain family proteins, and consequently reduces both mRNA nuclear export and cytoplasmic translation, without significantly altering mRNA steady-state amount.

## Reduced m⁶A site availability in predicted c.4444 C > A variant-specific mRNA structure

mRNAs can exhibit partially unfolded RNA structures during translation, and m⁶A modification can impact coding sequence structure and translation[45,50–52]. As local RNA structure can impact m⁶A modification, we propose that the conformation of WT *EPRS1* mRNA permits m⁶A modification in terminal exons 31-32, but C-to-A substitution at nucleotide 4444 induces a conformational switch that reduces or

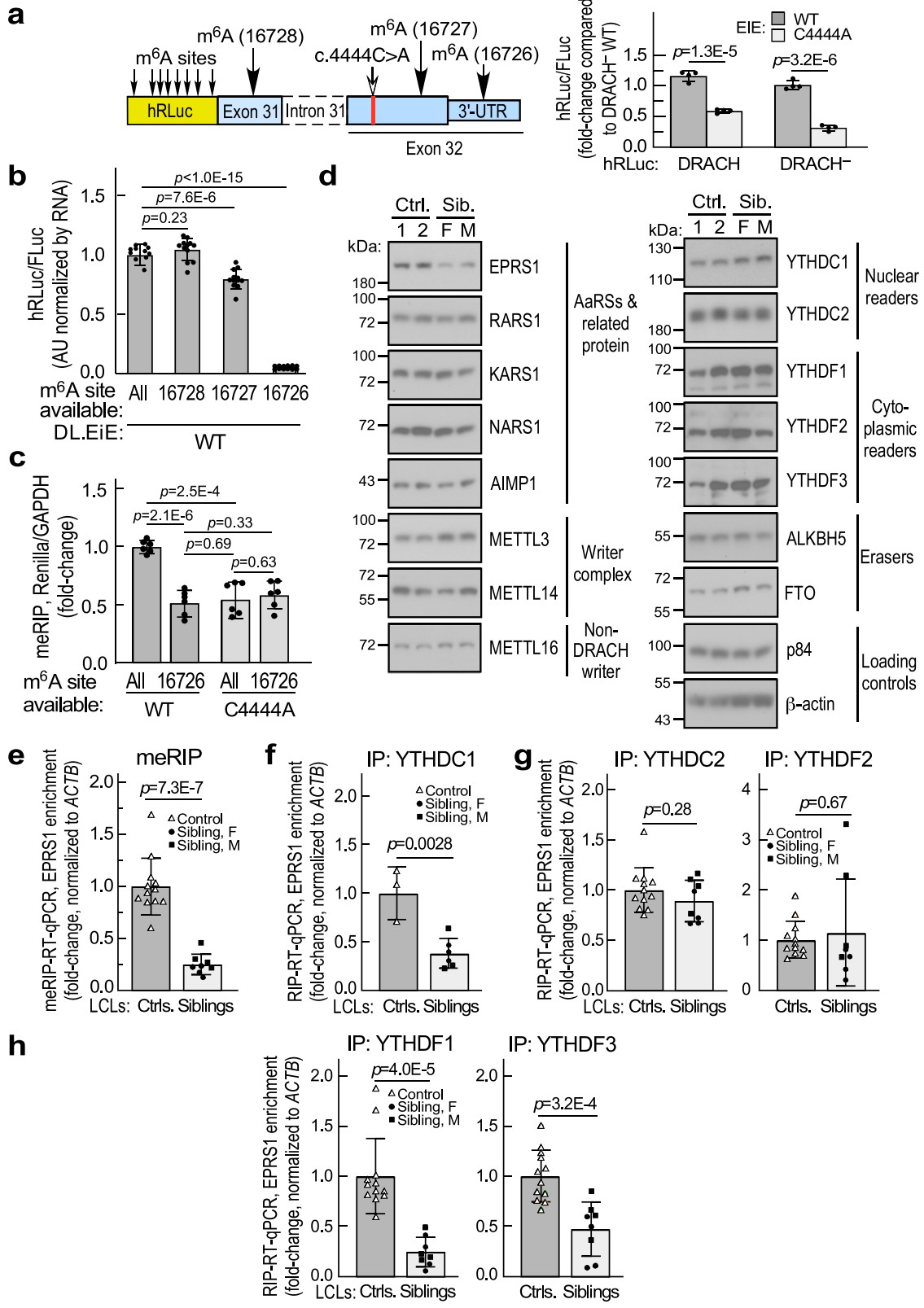

prevents modification. Minimum energy folding (RNAStructure 6.4[53]) of the local WT sequence, i.e., exon 31 and exon 32 up to the stop codon without the intervening intron, indicates C4444 is in a 5-bp stem including four sequential G·C base pairs (Fig. 5a, top); inclusion of the intron does not alter the folding structure in this region. The stem potentially facilitates a conformation in which the critical modified adenosine residues (16727, 16728) are unpaired within loop structures,

consistent with susceptibility to m6A modification[54]. The folding analysis indicates the c.4444 C > A variant disturbs the 5-bp stem, and the alternative structure is stabilized by two separate stems encompassing the m6A sites surrounding the variant site (Fig. 5a, bottom). The critical adenosine residues are within the base-paired stems, and thus less susceptible to m6A modification by the METTL3 complex[54]. The role of the 5-bp stem on reporter expression was explored by mutagenesis in

**Fig. 4 | Effect of C4444A mutation on m⁶A modification of *EPRS1* reporter expression and processing of endogenous *EPRS1* mRNA in patient LCLs.**
**a** Elimination of DRACH sequences in hRLuc reporter (DRACH⁻, left) enhances inhibition of expression by C4444A mutation (right). Mean ± SD, *n* = 4 biological replicates; *p* values are from unpaired two-tailed *t*-test. **b** Normalized expression of DRACH⁻ hRLuc reporter in HEK293T cells following pairwise mutation of m⁶A sites, pooled results for 293T and U87-MG cells that are shown individually in Supplementary Fig. 9a. Mean ± SD, *n* = 12 biological replicates (6 for each cell line); *p* values are from unpaired two-tailed *t*-test. **c** Detection of m⁶A modification of reporters by anti-m⁶A RNA immunoprecipitation (RIP), pooled results for 293T and U87-MG cells that are shown individually in Supplementary Fig. 9b. Mean ± SD, *n* = 6 biological

replicates (4 for 293T, 2 for U87-MG); *p* values are from unpaired two-tailed *t*-test. **d** Immunoblots showing steady-state levels of m6A readers, writers, and erasers in control and sibling LCLs. **e** Detection of m⁶A modification WT and C4444A variant *EPRS1* mRNA in patient and control LCLs by anti-m6A RIP-RT-qPCR. Detection of *EPRS1* mRNA binding to YTHDC1 (**f**), YTHDC2 and YTHDF2 (**g**), and YTHDF1 and YTHDF3 (**h**) by RIP-RT-qPCR in patient and control LCLs. For (**e**, **g**, **h**) Mean ± SD, *n* = 12 pooled biological replicates for control LCLs, *n* = 8 pooled female and male biological replicates for sibling LCLs. For (**f**), Mean ± SD, *n* = 3 pooled biological replicates for control LCLs, *n* = 6 pooled female and male biological replicates for sibling LCLs. *p* values are from unpaired two-tailed *t* test. Source data are provided in figshare repository [https://doi.org/10.6084/m9.figshare.25607931].

the DRACH⁻ hRLuc background. Mutation of C4444 to A, i.e., generation of the patients' variant, reduced hRLuc expression by about half, but restoration of the predicted stem by G4347U mutation in the opposing strand to generate an A-U base pair restored reporter expression (Fig. 5b). Mutation of G4347 to non-complementary nucleotides C or A did not restore reporter activity, supporting the predicted local structure and the critical role of the 5-bp stem. The complementary mutations retained the amino acids encoded by the c.4444 C > A (p.Pro1482Thr) reporter, indicating that amino acid sequence in the HLD variant is not responsible for the reduced expression. Three stem G-C pairs were reversed to C-G (including one encompassing the variant), one also was exchanged for an A-U pair, and all exhibited near-WT reporter activity, providing further evidence for the predicted stem (Supplementary Fig. 10a). The 5-bp stem is highly conserved in placental mammals providing evolutionary evidence for the importance of the 5-bp stem (Supplementary Fig. 10b, top). Two wobble base cytosines (for isoleucine and proline) in G-C bps appear as uridine in several mammals, e.g., mice, maintaining a G-U bp, consistent with a significant role of the stem. Interestingly, unlike the Pro1482 codon in one strand of the proposed stem, other proline codons in the window exhibit a different wobble base and higher degeneracy. Formation of the stem is a relatively recent event as it is not conserved in most other vertebrates, except in some reptilia (Supplementary Fig. 10b, bottom).

In an orthogonal approach, masking of the m⁶A sites by the putative stems in the variant mRNA was tested by mutations designed to disrupt the stems and expose cryptic m⁶A sites. Disruption of the stem surrounding the 16727 m⁶A site, by a series of synonymous and non-synonymous mutations (Gln1444Ile, Ile1445Gln, and Pro1446Pro) that minimally altered primary protein sequence, restored expression of the reporter bearing the C4444A mutation (Fig. 5c). Likewise, mutation of residues in the stem surrounding the 16728 m⁶A site (Ile1451Ile and Ile1481His), restored, and possibly exacerbated, reporter expression. These results are consistent with the previous finding that m⁶A modification of a single site is sufficient for EPRS1 expression. Together, the effects of the mutations suggest a mechanism in which the c.4444 C > A variant reduces accessibility of the m⁶A site to the methyltransferase, and putatively, RNA structure, not linear sequence, is the critical determinant of reporter expression.

The terminal two exons in *EPRS1* mRNA contains thirteen DRACH sequences, including three potential polymethylated regions in which multiple m⁶A sites are within a - 20 to 25-nt window, each including site 16728, 16727, or 16726 (Supplementary Fig. 11a). Site-specific alteration of m⁶A modification in patient LCLs compared to control LCLs was interrogated in twelve DRACH sequences in this region by SELECT-qPCR (single-base elongation- and ligation-based qPCR)[55]. In this method, the extension and ligation steps joining antisense probes flanking an adenosine residue are hindered by *N*⁶-methylation of the residue, resulting in reduced amounts of linked template from modified substrates[55,56]. Subsequent qPCR amplification, with primers complementary to terminal adapter sequences of the ligated probe-pair template identical for all sites, reveals altered methylation at target base. SELECT-qPCR revealed hypomethylated DRACH sites in the

terminal exons of *EPRS1* mRNA in patient LCLs (Fig. 5d and Supplementary Fig. 11b). Specifically, methylation at sites 16728 (A4355) and 16727 (A4464) in patient LCLs were lower by ~58% and ~29%, respectively (Fig. 5d, 1st and 2nd panels), whereas methylation at site 16726 (A4690) was unaltered (Fig. 5d, 3rd panel). As a control, m⁶A methylation at a validated site on 28S rRNA[55] was unaffected in patients LCLs (Fig. 5d, 4th panel). As additional controls, SELECT-qPCR targeted at non-DRACH adenosine residues near sites 16728 and 16726, i.e., A4349 and A4469, respectively, ruled out altered probe-pair accessibility in denatured template RNAs (Fig. 5d, 5th and 6th panels). *EPRS1, GAPDH,* and *ACTB* mRNA input levels were similar in control and patient LCLs (Fig. 5d, 7th panel and Supplementary Fig. 11b, bottom, right-most 2 panels). An ~40% reduction in methylation was observed at A4704 and A4716 near the terminus of the *EPRS1* 3'UTR. Hypomethylation at A4614 and A4666 in patient LCLs was not statistically significant due to large variation in the control LCLs. Modest hypermethylation at A4404 was seen in patient cells. We failed to detect a specific SELECT-qPCR signal at the terminal (13th site) in *EPRS1* mRNA owing to its proximity to the poly-A site, and consequently a low specificity poly-T up-probe that resulted in a multi-species melt-curve upon qPCR (sequences provided in Supplementary Table 4).

To verify target specificity of relevant probe-pairs SELECT-qPCR was performed on total RNA isolated from control LCLs subjected to *EPRS1* knockdown (Supplementary Fig. 12a, b). Higher Cₜ in *EPRS1* knockdown cells compared to LCLs nucleofected with non-targeting (NT) siRNA suggested probe-pair specificity, with the single exception of site A4614 which again exhibited substantial signal variation between controls. To further validate m⁶A methylation at specific adenosines, total RNA from FTO demethylase-treated control LCLs was subjected to SELECT-qPCR and compared to RNAs pre-quenched with EDTA to inactivate FTO. FTO treatment decreased Cₜ, validating SELECT-qPCR signals at m⁶A modification-specific *EPRS1* sites, as well as in the 28S rRNA methylation control; as expected, the signal at non-DRACH control site A4349 was not inhibited by EDTA (Supplementary Fig. 13a, b). Hypomethylation of sites 16728 and 16727, but not 16726, corroborated the reporter and MeRIP-RT-qPCR assays. In view of the unexpected reduction of methylation at sites A4704 and A4716 in the 3'UTR, we investigated the role of these sites in expression of the *EPRS1* WT reporter (Supplementary Fig. 14a). As before simultaneous disruption of the 16728 and 16727 sites (corresponding to A4355/A4464) abrogated reporter activity; however, mutation of the A4704/A4716 pair was without effect (Supplementary Fig. 14b). Together these results suggest that C4444A-directed alteration of RNA structure at sites 16728 and 16727 are sufficient to induce the observed change in EPRS1 expression.

## mRNA-targeted rescue of defective expression of c.4444 C > A variant

The structure-based inhibition of m⁶A modification suggests that non-genetic intervention might also increase availability and modification of m⁶A sites. Antisense phosphorodiamidate morpholine oligonucleotides (PMOs) were applied as steric blocks to disrupt base-paired regions containing m⁶A sites in the variant mRNA (Fig. 6a). PMOs were

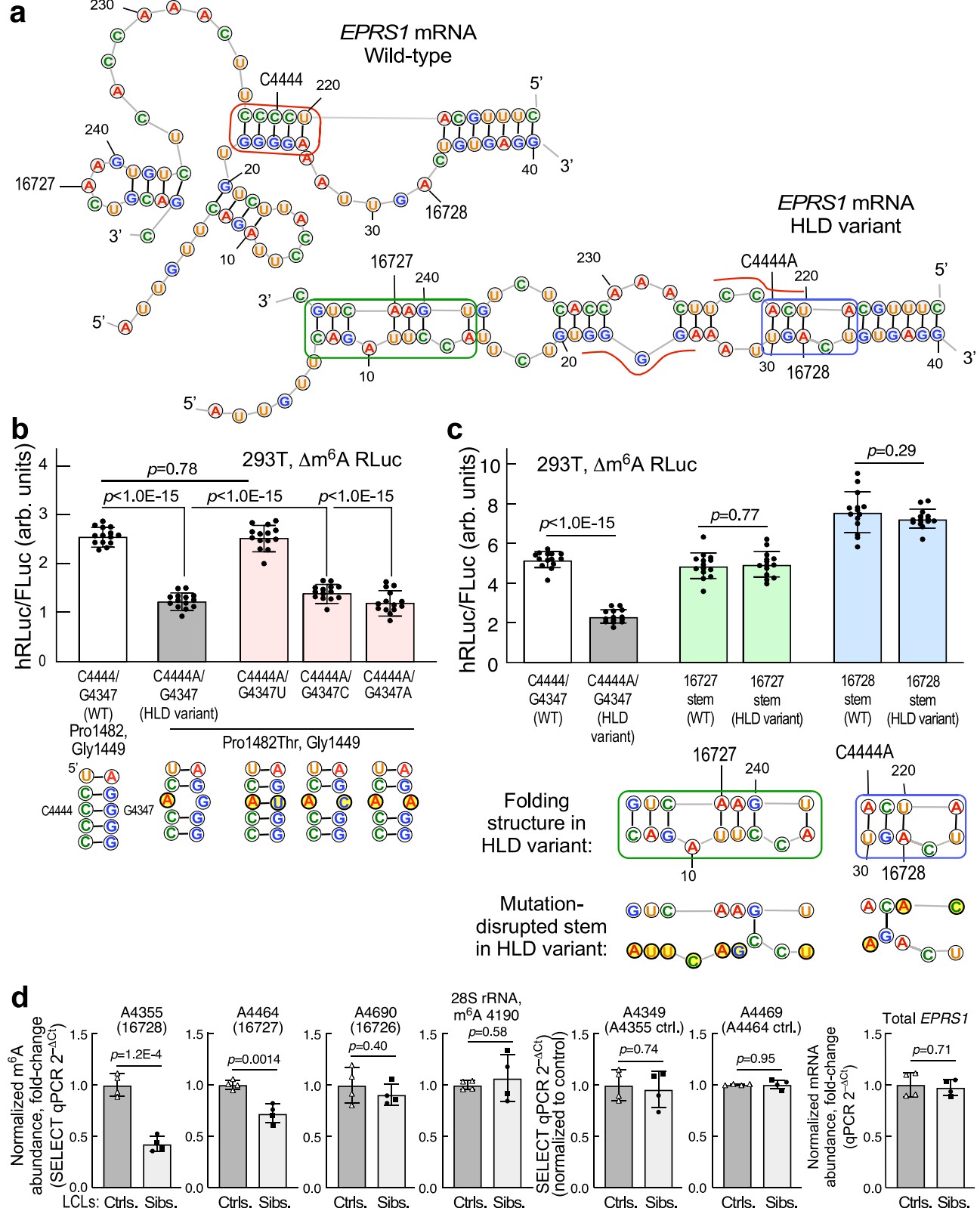

**Fig. 5 | Identification of m⁶A sites defective in hRLuc reporter bearing c.4444 C > A variant in *EPRS1* c.4444 C > A mRNA. a** Folding model of exons 31 and 32 of WT (top) and C4444A variant (bottom) *EPRS1* mRNA. Stems near C4444 site (red), near 16727 m⁶A site (green), and 16728 m⁶A site (blue) are highlighted. **b** Effect of mutations in the 5-bp stem surrounding C4444A site on hRLuc reporter expression in 293 T cells. Mean ± SD, *n* = 14; *p* values are from unpaired two-tailed *t*-test. **c** Effect of mutations in the stems surrounding 16727 and 16728 m⁶A sites on hRLuc reporter expression in 293 T cells. Mean ± SD, *n* = 14; *p* values

are from unpaired two-tailed *t*-test. **d** Differential SELECT-qPCR signals at specific DRACH sequence adenosine residues in *EPRS1* (first three panels from left), 28S rRNA (4th panel), and non-DRACH adenosine residues (5th and 6th panels) in control and patient LCLs. *EPRS1* mRNA levels by RT-qPCR of total RNA from LCLs (7th panel). Mean ± SD, *n* = 4 pooled biological replicates for control and sibling LCLs; *p* values are from unpaired two-tailed *t*-test. Source data are provided in figshare repository [https://doi.org/10.6084/m9.figshare.25607931].

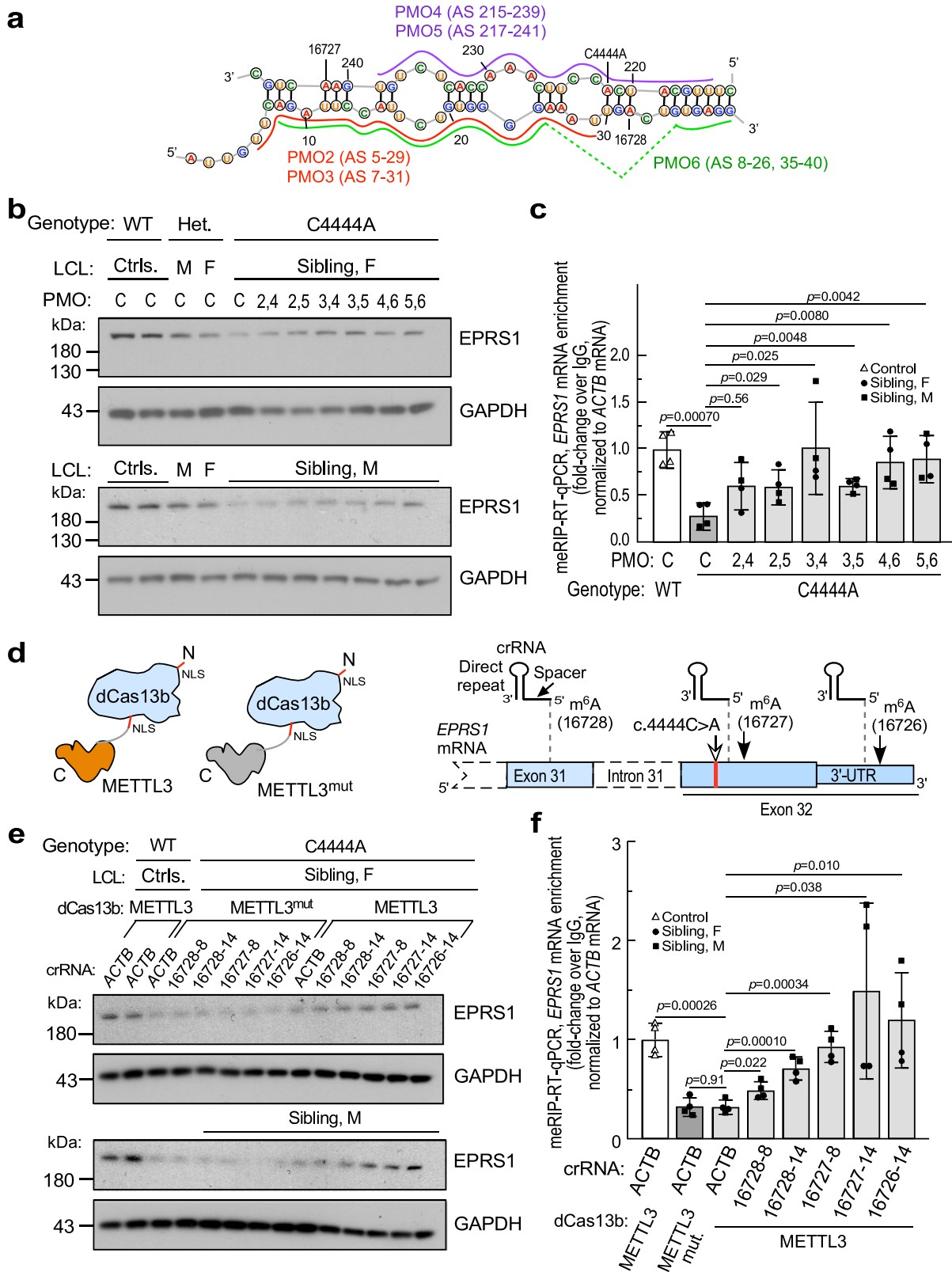

applied to patient LCLs, and lysate EPRS1 determined by immunoblot. All test PMOs induced EPRS1 expression compared to a control PMO; induced expression was higher in LCLs from the female compared to the male patient (Supplementary Fig. 15a). To increase expression, PMOs were added in pairwise combinations targeting both strands of the c.4444 C > A variant-specific structure. EPRS1 expression was increased, particularly with the PMO5/PMO6 pair to nearly the level in

the heterozygous parental LCLs (Fig. 6b). Increased protein expression was not accompanied by increased *EPRS1* mRNA consistent with post-transcriptional regulation (Supplementary Fig. 15b), Enhanced m⁶A modification of *EPRS1* mRNA by PMOs was shown by meRIP-qPCR (Fig. 6c).

In a second approach, rescue of defective m⁶A-modification in variant *EPRS1* mRNA was attempted using targeted RNA methylation

**Fig. 6 | Targeted rescue of expression of c.4444 C > A *EPRS1* variant in patient LCLs. a** Folding model of C4444A variant *EPRS1* mRNA exons 31-32 with antisense PMOs highlighted. **b** LCLs from controls, parents, and female (top) and male (bottom) siblings were incubated with control or selected pairs of antisense PMOs every 2 d for 6 d, left for 1 d, harvested after a total of 7 d, and EPRS1 expression determined by immunoblot. **c** Effect of PMOs on m⁶A modification of EPRS1 mRNA in WT and patient LCLs was determined by meRIP-qPCR. Mean ± SD, *n* = 4 pooled male and female siblings; *p* values are from unpaired two-tailed *t*-test. **d** Application of targeted RNA methylation (TRM) system. dCas13b with an

N-terminus NLS is ligated to METTL3 or inactive METTL3^mut (left). crRNAs with direct repeats were targeted 8 or 14 nt upstream of the three m⁶A sites in *EPRS1* mRNA exons 31-32 (right). **e** LCLs from controls and female (top) and male (bottom) siblings were nucleofected with crRNAs and chimeric dCas13b linked to METTL3 or METTL3^mut for 96 h, and EPRS1 expression determined by immunoblot. **f** Effect of TRM on m⁶A modification of *EPRS1* mRNA in WT and patient LCLs was determined by meRIP-RT-qPCR. Mean ± SD, *n* = 4 pooled male and female siblings; *p* values are from unpaired two-tailed *t*-test. Source data are provided in figshare repository [https://doi.org/10.6084/m9.figshare.25607931].

(TRM)[57]. Specifically, catalytically-dead Cas13b (dCas13b) bearing nuclear localization signals (NLS), and fused to truncated METTL3 methyltransferase was employed; inactive METTL3 mutant (METTL3^mut) served as a specificity control[57,58] (Fig. 6d, left). CRISPR RNAs (crRNAs) complementary to regions upstream of the m⁶A site and targeted by the dCas13b-METTL3 chimera served as guides. crRNAs were generated to target protospacer sequences 8 or 14 nucleotides upstream of the three m⁶A sites in exons 31 and 32 (Fig. 6d, right); crRNA targeting β-actin mRNA (*ACTB*) served as a control. dCas13b-METTL3 chimeras and crRNAs were nucleofected into control LCLs and affected sibling LCLs, and cells grown for 4 days. Nucleofection of dCas13b-METTL3 with crRNAs targeting each m⁶A site showed at least partial rescue of EPRS1; crRNAs targeting the two 3′-most sites were most effective in restoring in both patient LCLs, approaching that of control LCLs. As a control, crRNAs co-transfected with dCas13b-METTL3^mut were ineffective (Fig. 6e), demonstrating m⁶A writer-dependent rescue. These results are consistent with a role for deficient m⁶A modification in reduced expression of c.4444 C > A variant *EPRS1* mRNA. In parallel experiments, enhanced m⁶A modification of c.4444 C > A variant *EPRS1* mRNA roughly corresponding to the stimulation of expression was shown by meRIP-qPCR (Fig. 6f). Increased protein expression was not accompanied by increased *EPRS1* mRNA (Supplementary Fig. 15c). The influence of forced m⁶A modification at the 16726 site on EPRS1 expression was unexpected. Possibly, binding of m⁶A readers at sites distinct from the CDS sites near 16727/8 is induced. Although site 16726 modification state is not altered by C4444A mutation in the siblings, forced modification remains a viable therapeutic target.

To further verify that simultaneous disruption of methylation at sites 16728 and 16727 accounts for the observed reduction of EPRS1 expression, we explored the effect of site-specific demethylation. Guide RNAs targeting these sites singly or in combination were nucleofected into control LCLs with a nucleus-targeted, catalytically dead RfxCas13d (dCasRx)-ALKBH5 demethylase fusion protein[59–61] (Supplementary Fig. 16a). Guiding the demethylase to either site 16728 or 16727 did not reduce EPRS1 expression, but when targeted together decreased EPRS1 levels by about 50%−reaching 70–80% in several replicates (Supplementary Fig. 16b), close to the ~75% decrease in EPRS1 expression in sibling LCLs shown above.

### Widespread m⁶A site-distal single-nucleotide variants predicted to alter DRACH site accessibility

The prospect of additional single-nucleotide variants (SNV) that bury or expose distal DRACH sites through altered base-pairing in local RNA structure was investigated. SNVs in the ClinVar database of health status-associated genomic variations were cross-referenced with validated m⁶A sites in the RMVar database of RNA base modifications for all NCBI Refseq transcripts. Analysis was confined to the neighborhood of the CDS-terminal hotspot region for m⁶A modifications[29,62], by using 100 nucleotides of the 3′UTR following the stop-codon, and the preceding up to 150 nucleotides restricted to the last two exons in the coding region. Energy-minimized, predicted secondary structures of the wild-type and ClinVar SNV-containing mRNAs were calculated with RNAfold, and changes in predicted base-pairing at DRACH sites were determined. 117 hits in 54 genes and 87 ClinVar SNVs were identified

as candidate m⁶A-distal (m⁶Ad) SNVs (Supplementary Table 3 and index.html file in Supplementary Information); notably, multiple ClinVar SNVs can alter m⁶A-site accessibility of a given gene, while multiple but not all transcripts of a gene can be affected by a single ClinVar SNV. Notably, the EPRS1 c.4444 C > A; p.Pro1482Thr missense SNV (rs930995541) is absent from ClinVar database, and therefore not included in the list of predicted m⁶Ad-SNVs. Importantly, twenty candidates encompassing 11 genes and 14 ClinVar SNVs were from synonymous m⁶Ad-SNVs, potentially representing a new class of silent mutations that can alter gene expression and pathogenicity by altering m⁶A-site accessibility. DRACH-motif nucleotide base-pairing was scored as a measure of m⁶A site-accessibility[54]. As one example, a m⁶Ad-SNV in Von Hippel-Lindau mRNA (*VHL*, ClinVar ID 2224) is predicted to free two DRACH sites base-paired in the reference mRNA, and block accessibility of a third site (Fig. 7a). Similarly, a m⁶Ad-SNV in tuberous sclerosis complex 2 mRNA (*TSC2*, ClinVar ID 468159) predicts increased availability of three DRACH sites base-paired in the reference mRNA (Fig. 7b). PANTHER gene ontology analysis (Supplementary Fig. 17a), and DAVID functional annotation clustering of genes predicted to contain m⁶Ad-SNVs, reveal that these SNVs might impinge on critical biological processes and pathways, such as heart development and DNA damage (Supplementary Fig. 17b). These newly predicted m⁶Ad-SNVs, by altering accessibility of the methyltransferase to distant m⁶A sites, contrast with the established direct-acting m⁶A-SNPs that alter DRACH or near-DRACH sequences to inactivate existing m⁶A sites or generate new ones, respectively[63]. Importantly, as we have shown for the HLD-causing m⁶Ad-SNV in *EPRS1* mRNA, pathologies induced by the newly revealed m⁶Ad-SNVs might be correctible by RNA-based therapeutics.

## Discussion

There is emerging interest in the role of mRNA m⁶A modification in human pathology[64,65]. In some cases, the expression of an enzyme constituent of the m⁶A-modification pathway is altered. For example, FTO (fat mass and obesity-associated protein) the major m⁶A eraser, i.e., demethylase, is highly expressed in multiple acute myeloid leukemias, thereby enhancing oncogene-mediated cell transformation and leukemogenesis[66]. Pathogenic genetic variants that perturb m⁶A modification have been identified in two categories: (1) variants in m⁶A pathway enzymes, i.e., in writers, readers, and erasers, or (2) variants in DRACH or near-DRACH sequences themselves. In the first category, homozygous variants in the m⁶A reader *YTHDC2* in three women are associated with early-onset primary ovarian insufficiency[67], a clinical feature also seen in Sibling 2 in our study. Also, a pathogenic variant in FTO has been described in multiple members of a consanguineous Palestinian Arab family responsible for an autosomal-recessive lethal syndrome, and death before 30 months[68]. The variant inactivated DNA demethylation activity, but possibly demethylation of m⁶-modified mRNA was also inactive. The second category has received substantial recent attention in the form of SNPs inducing gain- or loss-of-function mutation of m⁶A sites, termed m⁶A-SNPs[69]. Two variants that generate pathologic m⁶A sites have been reported, both involving tumor suppressors. A G > A variant of the tumor suppressor p53 introduces an Arg273His missense substitution that promotes m⁶A modification of the mutant codon and increases expression, possibly by enhanced

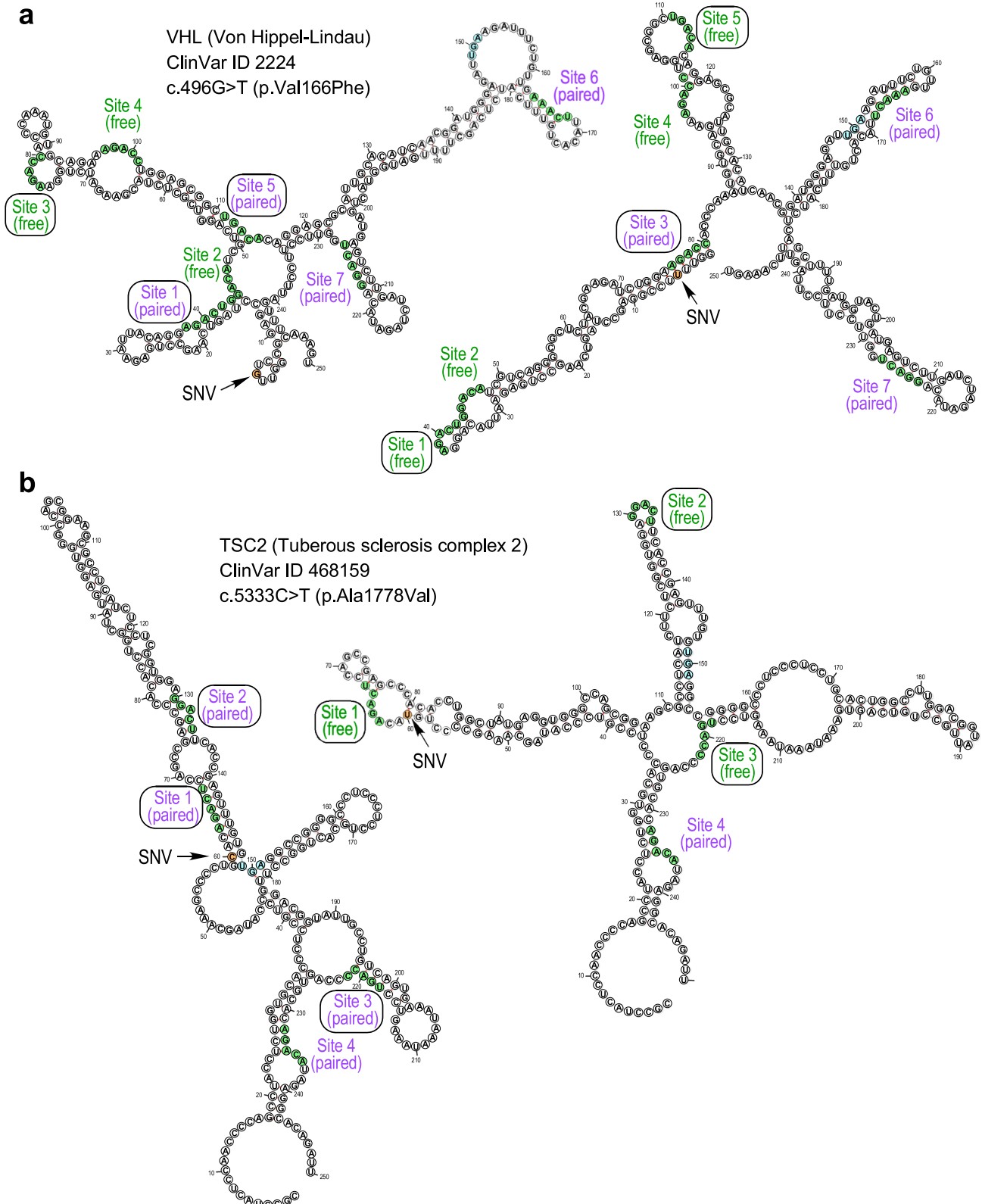

**Fig. 7 | Predicted changes in base-pairing of m⁶A-sites driven by health status-associated single nucleotide variants (SNVs).** RNAfold analysis of a selected 250-nt RNA sequence (see Methods for details) in *VHL* (**a**) and *TSC2* (**b**) transcripts, without (left) or with (right) a ClinVar-curated SNV. Highlighted are DRACH site (green), m⁶Ad-SNV (orange), and stop codon (light blue) nucleotides. DRACH sites are enumerated 5′ to 3′ and labeled in green if 3 or more nucleotides are unpaired or in magenta if 3 or more nucleotides are paired. DRACH sites predicted to be altered by m⁶Ad-SNV are boxed on both reference and alternate sequences.

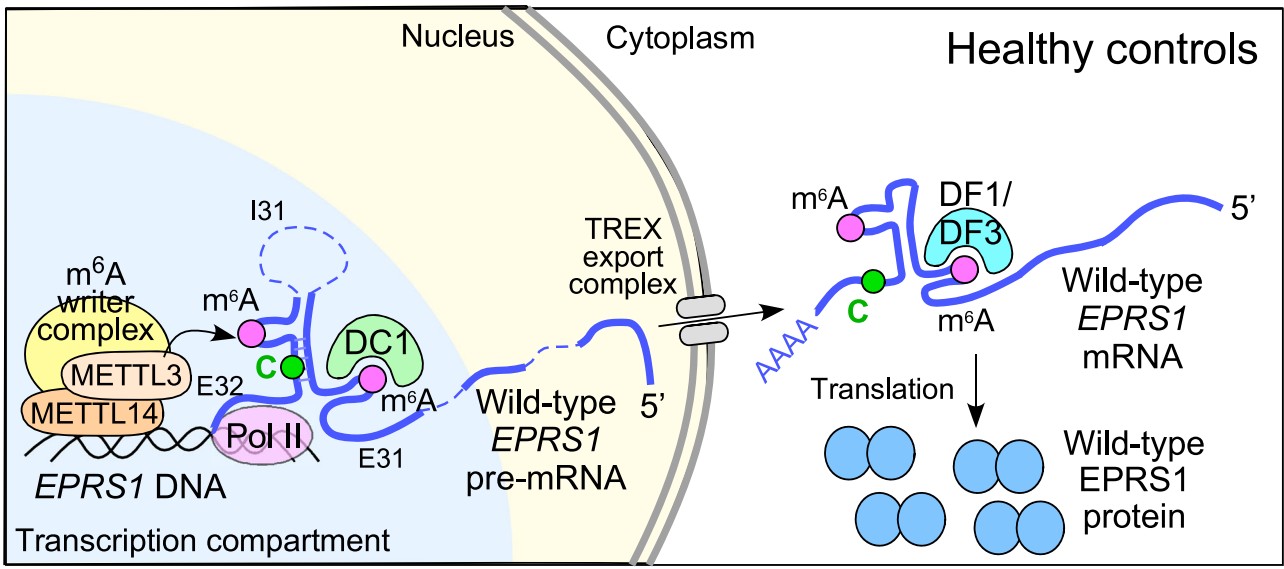

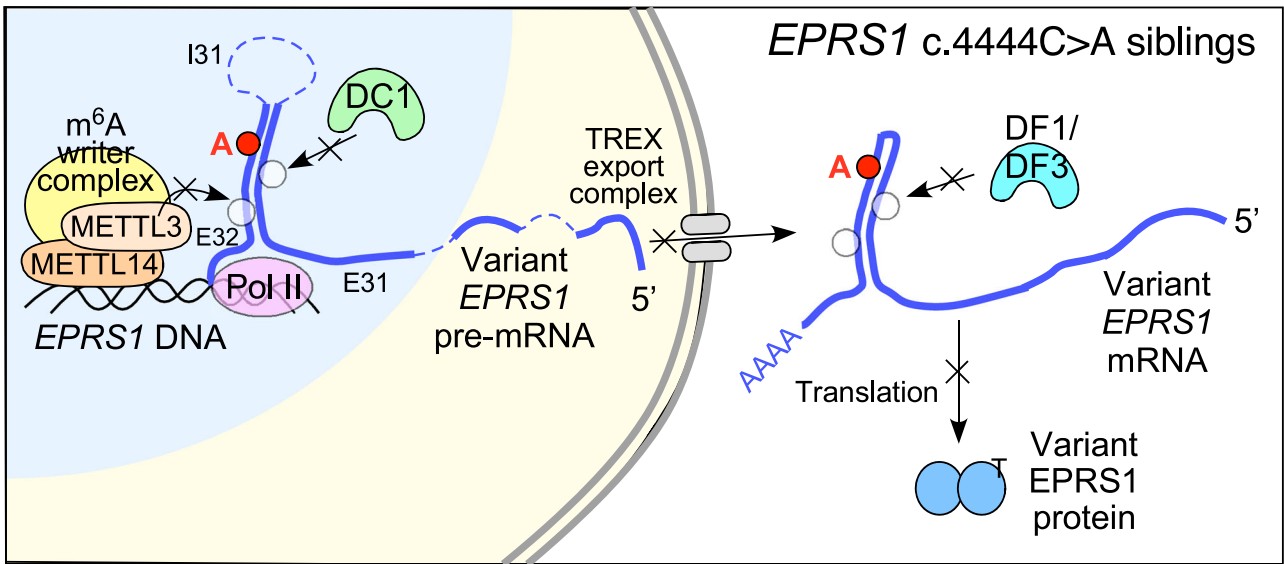

**Fig. 8 | Schematic of defective processing of *EPRS1* mRNA bearing c.4444 C > A variant.** Processing of wild-type (top) and C4444A HLD variant *EPRS1* mRNA (bottom). Occupied and unoccupied m⁶A sites are indicated by magenta and open circles, respectively. Abbvs.: DC1 YTHDC1, DF1 YTHDF1, DF3 YTHDF3, E31 exon 31, E32 exon 32, I31 intron following exon 31.

pre-mRNA splicing; the variant exhibits enhanced drug resistance[70]. In a second study, a G > A variant of *ANKLE1*, a suppressor of colorectal cancer (CRC), induced m⁶A modification, increasing ANKLE1 expression and reducing CRC risk[71]. Several disease-associated m⁶A-SNPs that repress methylation have been identified by combined analysis of GWAS and eQTL data[69]. For example, a SNP in acyl-CoA synthetase medium chain family member 5 (*ACSM5*), a candidate gene for thyroid cancer, reduces m⁶A modification and ACSM5 expression in thyroid cancer tissues, and is associated with the poor prognosis[63].

Our experiments reveal a third category of pathogenic genetic variants affecting m⁶A modification. In this case, the variant does not disrupt the target adenosine residue, nor the surrounding consensus sequence, but rather is located distal to the DRACH site (Fig. 8). We have observed that optimal expression of wild-type EPRS1 requires mRNA conformation-dependent m⁶A modification. Moreover, mutagenesis experiments indicate a requirement of an intact 5-bp putative stem bridging exons 31 and 32 for optimal methylation and gene expression. Importantly, the patient SNV at C4444 is centrally located in the 5-bp stem, thereby disrupting it and reducing modification at distant m⁶A sites, likely by a cis-allosteric alteration of mRNA conformation that reduces accessibility of otherwise modifiable m⁶A sites. This mechanism is supported by the finding that non-catalytic PMOs targeting the regions surrounding the inaccessible m⁶A sites in variant mRNA increase EPRS1 expression and mRNA methylation, likely by increasing accessibility to the methyltransferase and restoring gene expression. Upon probing differential m⁶A methylation in cells from patients and control subjects, hypomethylation in four DRACH sequences in the last two exons of EPRS1 was observed - two C4444A flanking sites in the CDS and two in the 3'UTR. Multiple lines of evidence, including reporter-based DRACH site mutagenesis assays, MeRIP-RT-qPCR, reporter-based DRACH-site unmasking assays, and site-targeted demethylation, supported the finding that loss of m⁶A methylation at sites 16728 (A4355) and 16727 (A4464) is sufficient to reduce EPRS1 expression in patient cells by regulating mRNA export and translation. Dual functions of polymethylated m⁶A regions have been described. For example, the ~250 nucleotide coding region instability determinant (CRD) in the *MYC* mRNA CDS, containing polymethylated m⁶A regions in its sequence, binds IGF2BP1 and regulates both mRNA

translation and stability[72,73]. Our results suggest a structural poly-methylated region formed by the *EPRS1* CDS 3'-end, containing non-contiguous sites 16728 and 16727, that is disrupted by SNV C4444A. The unexpected observation that two 3'UTR sites, A4704 and A4716, are also hypomethylated in patient LCLs, is intriguing. Pairwise mutation of these residues does not phenocopy the reduced EPRS1 reporter expression observed with the 16728/16727 pair. Possibly, hypomethylation of these sites, although not etiological, is related to loss of EPRS1 expression itself. For example, EPRS1 (misannotated as glutaminyl-tRNA synthetase owing to cDNA sequence similarity) was shown to bind its own 3'UTR, but not the upstream CDS[74]. It remains to be determined if loss of EPRS1 alters m[6]A modification of 3'UTR DRACH sites. Finally, although the C4444A variant described here is non-synonymous, the altered amino acid sequence does not appear to be critical for the observed regulation of EPRS1 expression. This is supported by examples where synonymous variants are etiologic agents of disease, e.g., cancer-derived synonymous point mutations in *p53* mRNA reduce p53 activity through post-transcriptional control[75]. Similarly, synonymous variants might be found that alter distant m[6]A modification in a structure-dependent manner, thereby providing an additional mechanism by which synonymous mutations dictate functional consequences beyond altering mRNA splicing, transcription and translation factor binding, mRNA stability, and translation elongation[76].

Also notable is the finding that the SNV in the open reading frame of *EPRS1* mRNA reduces both nuclear export and translation in patient LCLs, resulting in pathologically low levels of EPRS1 protein. The requirement for m[6]A modification of wild-type *EPRS1* mRNA for efficient expression of protein was not previously reported for any aminoacyl-tRNA synthetase. Moreover, EPRS1 is a constitutive, housekeeping protein, and mRNAs encoding such proteins, for example, ribosomal proteins, generally are de-enriched in m[6]A[35]. In addition to defective m[6]A modification of mutant *EPRS1* mRNA in patient-derived LCLs, defective m[6]A modification was observed in mutation-bearing reporters in embryonic kidney-derived HEK293T cells and glioma-derived U87-MG cells. The apparent lack of cell-type specificity is consistent with previous observations by others that m[6]A sites are generally constitutive with similar distributions in tissues and cell lines[35,36]. Additionally, m[6]A enzymes, including writers and readers are present in most tissues including major brain cells, such as neurons and neuroglia, the cell types likely to be adversely affected by dysregulated expression in HLD patients[65]. Importantly, the m[6]A epitranscriptome is implicated in oligodendrocyte maturation, CNS myelination, and brain development[77–79].

Four other patients with distinct bi-allelic pathogenic variants in *EPRS1* exhibiting childhood-onset HLD15 have been reported[11]. Two are homozygous for missense variants in the ProRS catalytic core domain; the other two are compound heterozygous, characterized by one allele with a missense variant in the ProRS catalytic core and a premature stop codon in the other allele. The best studied of these mutant forms is the c.3344 C > G variant which encodes a p.Pro1115Arg substitution in the catalytic core domain predicted to influence specific activity. Indeed, an ~30% decrease in charging activity by recombinant ProRS was shown as well as a 40% decrease in EPRS1 protein in patient fibroblasts. In vitro charging activity in lymphoblast lysates was determined in one of the patients and showed ~70% reduction compared to controls. These results, combined with our own, suggest a mechanism in which diminished total charging activity—whether due to protein amount or specific activity, or both—contributes to the pathologic defect.

The mechanistic link between reduced EPRS1 and CNS hypomyelination remains elusive. The obligate role of EPRS1, like all aaRSs, in interpretation of the genetic code during translation suggests that reduced protein synthesis is a major contributor to pathology. The undiminished total protein synthesis in our patients' LCLs argues against this mechanism. However, low levels of EPRS1 might specifically inhibit protein synthesis in critical cells particularly sensitive to tRNA charging activity, e.g., in myelinating oligodendrocytes. Consistent with tissue-selective responses to aminoacylation defects, fibroblasts from patients with compound heterozygous mutations in the GluRS region of EPRS1 exhibited normal growth rates despite severely compromised tRNA charging activity of recombinant protein in vitro[80]. Although genetic defects in two other cytosolic aaRSs, namely, DARS1 and RARS1, also cause HLD[10,81,82], defects in at least ten cytosolic aaRSs cause distinct neurologic disorders including encephalopathy, microcephaly, as well as peripheral neuropathy[1,11], suggesting that aaRS inhibition of protein synthesis is unlikely to be the principal etiology underlying HLD. Alternatively, the defect might be due a defective noncanonical function of EPRS1, possibly a CNS-specific function, unrelated to protein synthesis[22,23,83,84]. This mechanism would suggest that disease-causing variants in other aaRSs inhibit the same or related noncanonical activities—a concept not supported by current data. A possible clue is the observation that three of the variant genes that cause HLD, i.e., *EPRS1*, *DARS1*, and *RARS1*, encode constituents of the MSC which houses nine of the twenty cytosolic aaRSs. In addition, variants in two non-aaRS MSC constituents, i.e., *AIMP1* and *AIMP2*, also cause leukodystrophy, although this is likely to be secondary to a primary neurodegenerative process[12,13,85,86]. Thus, a dysfunctional MSC might contribute to HLD pathology. However; elucidation of this mechanism is hampered by the current lack of understanding of the critical function(s) of the MSC which appears to be unrelated to efficiency of protein synthesis[87].

Improved understanding of the pathophysiology of leukodystrophies, aided by state-of-the-art molecular technologies, has begun to guide potential therapeutic approaches, including drug design and gene therapy[88]. For example, an RNA-based therapeutic targeting proteolipid protein 1 (*PLP1*), the gene defective in Pelizaeus-Merzbacher disease (PMD), has been investigated. Administration of antisense oligonucleotides targeting *PLP1* restored myelination and motor function in the jimpy (Plp1(jp)) mouse model of severe PMD[89]. Likewise, a morpholino was used to correct aberrant splicing in a mouse bearing a *PLP1* exon 3 variant that in humans causes PMD, spastic paraplegia 2, and hypomyelination of early myelinating structures[90]. Lastly, administration of a locked nucleic acid antisense oligonucleotide targeting N-acetyltransferase 8-like (Nat8l), the enzyme that generates N-acetyl-L-aspartate, reverses ataxia in a mouse model of Canavan disease, a vacuolar leukodystrophy[91]. In the experiments described here, the mechanism of inhibition has significant implications regarding potential treatment to restore *EPRS1* mRNA processing. Diminished expression of the variant EPRS1 results from a single well-defined molecular event that suppresses both nuclear export and translation, namely, defective m[6]A modification of the transcript. Increasing gene expression using an antisense strategy against the coding sequence has been reported[92]. Rationally-designed, antisense PMOs were deployed to disrupt the predicted, variant-specific RNA secondary structure to reveal masked m[6]A sites. The modest rescue serves as a proof-of-principle for pre-clinical testing in animal models and, potentially, in eventual therapeutic application. Antisense PMOs, e.g., eteplirsen and golodirsen, are FDA-approved for clinical management of Duchenne muscular dystrophy[93,94] by altering splicing of mutant mRNA. Alternatively, tethering catalytically-inactive dCas13 to m[6]A writers, erasers, and readers for programmable, site-specific introduction of m[6]A has the potential to transform fundamental studies of RNA methylation, as well as clinical application[58]. Taking advantage of a targeted RNA methylation system[57], nucleus-localized dCas13b fused with truncated METTL3 methyltransferase was co-nucleofected with guide RNAs cognate to sites upstream of m[6]A sites in the two terminal exons of *EPRS1* mRNA. Highly effective restoration of EPRS1 expression in patient LCLs expands the therapeutic toolkit to ameliorate low EPRS1 expression associated with the c.4444 C > A variant.

RNA structural determinants of m⁶A modification sites have been proposed but a consensus structure is yet to be identified[29,95]; however, consensus sequences sequestered within RNA duplexes are known to be poor substrates for m⁶A modification[54,96]. The discovery of a SNV that allosterically blocks accessibility of distal RNA m⁶A sites, with subsequent altered protein expression, reveals a novel disease etiology: m⁶Ad-SNPs or -SNVs represent a third mechanism by which SNPs (or SNVs) alter m⁶A-dependent gene expression, in addition to SNPs dysregulating genes encoding m⁶A pathway proteins or within m⁶A-sites (m⁶A-SNPs) themselves. Genome-wide, bioinformatic prediction of a host of m⁶Ad-SNVs suggests this mechanism of disease might be widespread. In most cases, a single m⁶Ad-SNV alters the accessibility of multiple DRACH target sites within a transcript—both by masking existing sites or exposing new ones. Currently there is no experimental evidence that these predicted m⁶Ad-SNVs influence m⁶A modification of the transcript; however, limited biochemical evidence suggests ClinVarID 2224 in *VHL* might increase protein amount in 786-O renal carcinoma cells[97]. Although the *VHL* and *TSC2* variants illustrated here show non-synonymous, hydrophobic amino acid substitutions, many synonymous m⁶Ad-SNVs were predicted. The finding represents an additional mechanism by which silent mutations can influence gene expression and contribute to disease etiology. Validation of predicted m⁶Ad-SNV-driven changes in RNA structure, m⁶A modification, and gene expression are necessary to show the findings for *EPRS1* can be extended to other genes[98,99]. Analysis of m⁶Ad-SNV-altered RNA structures might reveal consensus structural determinants of m⁶A modification on mRNA. Importantly, the RNA-centric nature of the m⁶Ad-SNV mechanism of disease might generate opportunities for personalized, RNA-based therapeutic modalities targeting these disorders.

## Methods

### Patient recruitment and diagnosis

Both patients are followed in a specialized Neurogenetics Clinic at The Hospital for Sick Children (Toronto, Canada). Extensive genetic and non-genetic investigations were initially non-diagnostic (Supplementary Table 2). This study was approved by the Research Ethics Board of The Hospital for Sick Children (REB# 1000009004). The parents provided written informed consent to participate in this study, which included generation of cell lines from the study participants and publication of clinical details (including age, sex, medical centre where care was provided and rare diagnosis) and brain imaging. This is a case-level report where gender and sex were not significant determinants of outcome. We describe an autosomal recessive condition in two siblings, one male and one female. No sex or gender-based analyses were performed based on a sample size of one male and one female participant. Disaggregated data for sex is shown by symbols in all relevant figures and have been tabulated separately in figshare repository [https://doi.org/10.6084/m9.figshare.25607931].

For diagnosis, Quad-based whole-exome sequencing was performed in a Clinical Laboratory Improvement Amendments (CLIA)-approved laboratory (GeneDx; http://www.genedx.com/). This revealed that the affected siblings are homozygous for the missense variant c.4444 C > A; p.Pro1482Thr (rs930995541) in *EPRS1* (GenBank: NM_004446.2). Mean depth of sequence coverage was 98x, with 97.6% of the defined target region with read depth at least 10x, including 100% of the *EPRS1* coding region. This variant was not observed in >130,000 unrelated individuals in the Genome Aggregation Database (gnomad.broadinstitute.org/). The variant has been submitted to ClinVar post-study (ClinVar ID 3069175).

### Generation and culture of LCLs

Immortalized LCLs were generated from both affected patients, both carrier parents, and an unaffected control subject at The Centre for Applied Genomics, The Hospital for Sick Children (Toronto, Canada).

Peripheral blood (5–10 ml) was collected with anticoagulant (acid-citrate dextrose, ACD Solution A). Buffy coat containing peripheral blood mononuclear cells (PBMC) was separated by centrifugation on a Ficoll-Paque gradient (Cytiva). PBMCs were washed two times with RPMI-1640 (Wisent) and resuspended in 1 ml of RPMI 1640 containing 50% FBS (Cytiva), 2 mM L-glutamine or equivalent, 2.0 g/L glucose, 2.0 g/L sodium bicarbonate, without antibiotics. 0.5 to 1 ml Epstein Barr Virus (B95-8, ATCC CRL-1612, Accegen) and 1 μg/ml cyclosporine A were added, and cultures incubated at 37 °C, 5% $CO_2$. After 7–10 days the cells were transferred to a T25 flask by removing 1 ml of the cells and adding same volume of RPMI 1640 with 15% FBS (Cytiva). Twice weekly, flasks were visually examined for acidic pH (yellow media) and clumps (rosettes) of cells growing in suspension. The cells were fed or split at 1:1 ratio every 3–4 days. LCLs were sub-cultured at a seeding density of not less than $2 \times 10^5$ viable cells per ml. When cell density reached $0.8–1.0 \times 10^6$ cells per ml, the culture was split at not less than $2 \times 10^5$ cells per ml and cell stocks were cryopreserved. Additional race and ethnicity-matched control LCLs (Controls 1, 2) from unaffected subjects were provided by Dr. Charis Eng (Genomic Medicine Biorepository, Cleveland Clinic). Thawed LCLs were grown in RPMI 1640 (Lerner Research Core) supplemented with 15–20% fetal bovine serum (Gemini Bioscience) and a solution of penicillin/streptomycin and L-glutamine in a 37 °C incubator stabilized at 5% $CO_2$. Details of reagents and media are available in Supplementary Table 4.

### Cell lines and culture

HEK293T (CRL-3216) and U87-MG (HTB-14) cell lines were sourced from ATCC and 293F (11625019) was purchased from Thermo Fisher. HEK293F and HEK293T cells were maintained in high glucose DMEM and 10% heat-inactivated FBS (Gemini Bioscience). U87-MG cells were maintained in EMEM and 20% non-heat-inactivated FBS (Sigma). All media were supplemented with penicillin-streptomycin and 2 mM L-glutamine. ATCC stocks of U87-MG come from a different donor to the Uppsala stocks ([100] and Version 12 Table 2, misidentified cell lines, https://iclac.org/databases/cross-contaminations/). The ATCC stocks have CNS origin and is likely a bona fide human glioblastoma cell line, similar to Uppsala stocks, allowing its limited use in a role that is only secondary to patient-derived LCLs and HEK293T in this study. Since sourced from ATCC, these cells were not authenticated further.

### Molecular cloning, gene assembly, mutagenesis, and spacer design

Refer to Supplementary Table 4 for plasmids used. pCMV10-3X-FLAG-*EPRS1*[21] was used as template to generate Pro1482Thr mutant using QuikChange site-directed mutagenesis. hRLuc-EIE (exon31-intron 31-exon 32)-vector 3′UTR reporter plasmid (PF0720) was generated by cloning within the *Age*I and *Bam*HI sites of pEGFPC1 an NGS-validated gblock (IDT) containing a Kozak sequence (5′-CGCCACC-3′) upstream of humanized renilla luciferase (hRLuc, pGL4.70) without the stop codon, followed by the relevant *EPRS1* gene architecture (GeneID:2058, coordinates 77384-77905). A 199-nt stretch after the *Xba*I site in the pEGFPC1-derived SV40 polyA terminator sequence, that contains two polyA signals[101], was deleted using Q5 site-directed mutagenesis kit (NEB) to force utilization of the *EPRS1* 3′UTR polyA signal, and generate the PF0721 plasmid. The c.4444 C > A variant nucleotide (PF0722) and intron 31 deleted construct series (PF0701 and PF0702) were generated by site-directed mutagenesis in PF0721, and PF0721 and PF0722 were digested with *Xmn*I and *Bts*I.v2. A gBlock (IDT) containing *EPRS1* exon 30, without the first G nucleotide to maintain coding frame, (GeneID:2058, coordinates 77360-74437), rabbit β-globin intron 2 (GeneID:100009084, coordinates 707-1279), and cognate flanks were inserted using NEBuilder HiFi DNA Assembly (NEB) to generate PF0741 and PF0742. PF0721 and PF0741 were digested with *Bts*I.v2 and *Pst*I to delete *EPRS1* intron 31 and insert the chimeric intron (cI, GenBank:U47119.2, coordinates 857-989) to

generate PF0921 and PF0941 plasmids. The c.4444 C > A variant nucleotide was introduced by site-directed mutagenesis to generate PF0922 and PF0942 plasmids. PF0701/02/21/22 plasmids were digested with *Age*I, a gBlock (IDT) containing the chimeric intron was inserted by HiFi DNA Assembly, and *Age*I site was retained by design to generate PF0801/02/21/22 plasmids, respectively.

For DRACH-less hRLuc backbone, an NGS-validated gBlock containing hRLuc coding sequence with 8 DRACH sites disrupted (7 by synonymous mutations and one Thr184Ser mutation), and a segment of *EPRS1* exon 31, was inserted into *Ava*I and *Bts*I.v2 sites of PF0721 and PF0722 to generate DLRL0721 and DLRL0722, respectively. In the background of DLRL0721, *EPRS1* DRACH sites 16728, 16727, and 16726 were altered (by synonymous mutations for 16728 and 16727) pair-wise by site-directed mutagenesis. Complementary mutations to c.4444 C > A variant were generated in DLRL0722 by site-directed mutagenesis. Primary and complementary mutations to test the G4A:UC4 stem, as well unmasking mutations for sites 16728 and 16727, were generated in DLRL0721 and DLRL0722 by site-directed mutagenesis. ACTB spacer sequence and direct repeat sequence were deleted from pU6-PspCas13b-gRNA-Actb1216 (Addgene 155368) by *Bsa*AI and *Kpn*I digestion, and various *EPRS1* spacer sequences and direct repeat sequence were introduced. *EPRS1* spacer sequences were antisense to 30-nt protospacer regions either 8 or 14 bp upstream of methylated A nucleotide[57] of m6A methylation sites 16728, 16727, and 16726. A4704G and A4716G mutations were incorporated into DLRL0721 by site-directed mutagenesis. Guide RNAs for dCasRx-ALKBH5 were cloned in pXR003 utilizing *Nhe*I and *Eco*RI sites from chemically synthesized gBlocks (IDT). A non-targeting (NT) crRNA sequence was used as control[59]. For CasRx-ALKBH5 gRNA targeting any *EPRS1* DRACH site, $(N_{18})HCARD(N_7)$ was used as reverse complement of $(N_7)DRACH(N_{18})$ (Supplementary Table 4). Mutations and assemblies were validated by Sanger sequencing.

## Expression and purification of FLAG-tagged WT and Pro1482Thr mutant EPRS1

3XFLAG-EPRS1 WT and Pro1482Thr mutant were purified as described[21]. HEK293F cells (Thermo Fisher) were transiently transfected with each plasmid. Cells were harvested 72 h after transfection in TNE lysis buffer (50 mM Tris-HCl pH 7.4, 150 mM NaCl, 0.1 mM EDTA, 1% NP-40) containing HALT protease inhibitor (Thermo Fisher). Lysates were cleared by centrifugation at $16,000 \times g$ for 20 min. Pulldown was performed overnight with anti-DYKDDDDK G1 Affinity Resin (Genscript) at 4 °C. Beads were washed three times with TNE buffer, and proteins eluted with 3X-FLAG peptide (MilliporeSigma) according to the manufacturer's protocol. Protein concentrations were quantified, and fresh preparations were used for tRNA charging experiments.

## Determination of tRNA charging activity by recombinant EPRS1 and LCL lysates

LCLs were lysed in buffer containing 25 mM Tris-HCl pH 7.4, 150 mM NaCl, 1% NP-40, and 5% glycerol. After lysis, NP-40 concentration was adjusted to 0.2% with detergent-free buffer. tRNA aminoacylation activity was measured essentially as described[21]. Reactions were pre-equilibrated to 30 °C in 20 μL HEPES assay buffer (20 mM HEPES pH 8.0, 100 mM NaCl, 5 mM MgCl₂, 3 mM ATP, and 1 mM DTT) supplemented with 150 μM ¹⁴C-L-proline/L-glutamate/L-phenylalanine and 1 mg of total yeast tRNA. Reactions were initiated by addition of purified, 3X-FLAG-tagged WT or Pro1482Thr mutant EPRS1 (0.5 μM) or 30 μg of patient LCL lysate. Following incubation at 30 °C for 30 min, 15 μL aliquots were spotted on glass filters (Whatman GF/C™) pre-soaked with 5% trichloroacetic acid (TCA). Filters were washed three times with 1 mL of 5% TCA followed by 2 × 1 mL washes with 100% ethanol. Filters were dried in hybridization oven at 60 °C for 10 min, and radioactivity determined by liquid scintillation counting (TRICARB 1900TR, Perkin-Elmer).

## RNA isolation and RT-qPCR analysis

Total RNA was isolated from LCLs or 293 T cells using Qiagen RNeasy Mini Kit with DNase treatment. Equal amounts of total RNA were used for quantitative PCR with AgPath-ID One Step using gene-specific Taqman probe-primer sets (Thermo Fisher, refer to Supplementary Table 4 for probe-primer set details).

## 3′ RACE analysis of C4444A *EPRS1* mRNA

3′ RACE was done as described[102]. Briefly, total RNA was isolated from sibling LCLs using Trizol, and cDNA was generated by reverse transcription followed by PCR using SuperScript III One-Step RT-PCR System (Invitrogen). The 52-nt $(Q_0–Q_I–T)$ CCAGTGAGCAGAGTGACGA GGACTCGAGCTCAAGCTTTTTTTTTTTTTTTTT primer was used to reverse transcribe cellular mRNAs. Primers $Q_0$–CCAGTGAGCAG AGTGACG, $Q_I$–GAGGACTCGAGCTCAAGC and gene-specific primers were used in sequential amplifications to generate sequence-specific product.

## Polysome profiling

Isolation of ribosome-rich, translationally-active and ribosome-poor, inactive mRNA pools was done by sucrose gradient fractionation. Cycloheximide (100 μg/mL) was added to $10^6$ LCLs for 20 min, and cells collected by low-speed centrifugation and washed twice with cycloheximide-containing, ice-cold PBS. Cell pellets were suspended in 350 μL of lysis buffer (10 mM Tris pH 7.4, 5 mM MgCl₂, 100 mM KCl, 1% Triton X-100, 0.5% deoxycholate, 2 mM DTT, 100 μg/mL cycloheximide, and RNAse inhibitor) and incubated for 5 min on ice. The lysates were centrifuged at $15,000 \, g$ for 10 min and supernatants collected. RNase inhibitor (5 μL, 40 U/μL) and cycloheximide (100 μg/mL) were added to 50 ml each of freshly prepared 10% and 50% sucrose gradient solutions (20 mM HEPES pH 7.4, 100 mM KCl, 5 mM MgCl₂, and 2 mM DTT) just before use. Lysates were loaded onto the sucrose gradient and centrifuged at $220,000 \times g$ for 4 h, and 8 fractions of about 1 mL were collected and combined. Fractions containing light ribonucleoproteins, 40S, 60S, and 80S ribosomes formed the translationally-inactive pool, and heavy polysome fractions formed the translationally-active pool.

## Determination of total protein synthesis by LCLs

LCLs $(0.5 \times 10^6$ cells) were pre-incubated in methionine-free RPMI medium (Invitrogen) with dialyzed FBS (ThermoFisher) for 30 min, followed by addition of [³⁵S]Met/Cys (0.01 mCi, Perkin-Elmer) for 15 min at 37 °C. Labeled cells were lysed in RIPA buffer (Sigma) with protease and phosphatase inhibitors. Lysates from equal numbers of cells were resolved by SDS-PAGE. Gel was fixed in 40% methanol, 10% acetic acid, and processed by autoradiography.

## Purification of recombinant ProRS and determination of oligomeric state

Wild-type or P1482T mutant ProRS fragment of human EPRS1 (aa 930-1512) was subcloned in pTRC-HisB (Invitrogen) for N-terminal 6X-His tagging, and sequence verified. Recombinant protein was expressed in BL21 Codon-plus (DE3)RIPL (Agilent) strain as described[21,103]. Briefly, protein was induced with isopropyl β-D-1-thiogalactopyranoside (200 μM) at 37 °C, and cells harvested by centrifugation 4–6 h post-induction. The pellet was resuspended in purification buffer containing 50 mM Tris-HCl, pH 8.0, 100 mM NaCl, 10% glycerol, 1 mg/ml lysozyme, protease inhibitors, and 10 mM imidazole, and sonicated on ice for 20 min. Lysate was cleared by centrifugation at $26,000 \times g$ for 45 min, and purified using HisTrap HP column (GE Life Sciences, Pittsburgh, PA). Protein oligomeric state was determined using a Superdex 200 size-exclusion column (GE Life Sciences) pre-calibrated with Bio-Rad gel-filtration standards (Bio-Rad) in purification buffer on an NGC Chromatography System with ChromLab v5.0.2.11 (Bio-Rad). ProRS aminoacylation activity was confirmed as described[21].

## Analysis of EPRS1 localization in MSC

Wild-type and c.4444 C > A variant 3X-FLAG-*EPRS1* were transiently transfected in HEK293T cells in 10-cm cell culture dishes. After 72 h, cells were harvested and washed with ice-cold PBS. The pellet was re-suspended in RIPA buffer for 15 min at 4 °C in the presence of 1X protease inhibitor cocktail, and debris removed by centrifugation at $21,130 \times g$ for 15 min at 4 °C. Samples containing equal amounts of protein, quantified by BCA method, were subjected to overnight pulldown at 4 °C using Dynabeads protein-G (ThermoFisher) coupled to monoclonal anti-FLAG M2 antibody. The beads were washed three times with lysis buffer, and protein eluted with 3X FLAG peptide (MilliporeSigma) according to the manufacturer's protocol. Eluates were subjected to western blot analysis with anti-FLAG or anti-MSC resident aaRS-specific antibodies.

## Western blot analysis and antibodies

Samples harvested from cells or by immunoprecipitation and elution were mixed with RIPA buffer (Sigma), and subjected to SDS-PAGE. Antibodies against EPRS1, KARS1, LARS1, IARS1, AIMP3, SARS1, p84, METTL3, YTHDC1, YTHDC2, YTHDF1, YTHDF2, YTHDF3, GANP, NXF1, CRM1, IPMK, β-actin-HRP, FLAG, GAPDH-HRP and α-tubulin-HRP were used in immunoblot analysis (refer to Supplementary Table 4 for details on sources, dilutions, and validations).

## Determination of EPRS1 stability

Control, parental, and sibling LCLs ($0.5 \times 10^6$ cells) in 6-well plates were incubated with cycloheximide (100 μg/ml) in DMEM for 0, 12, and 24 h to inhibit protein synthesis. Lysates were subjected to SDS-PAGE and immunoblot analysis for α-tubulin and EPRS1, and quantitated by densitometry using ImageJ software.

## mRNA determination in nuclear and cytoplasmic fraction

LCLs (~5 million) were spun at 200 g for 5 min, pellet washed once with PBS, and re-suspended. Cells were spun again at $200 \times g$ for 5 min, and pellet subjected to cytoplasmic and nuclear RNA fractionation using PARIS kit (Life Technologies). RNA fractions were collected in 60 μl of kit elution solution and treated with Turbo DNase (Life Technologies) per manufacturer's protocol in 70 μl reaction volume. Following inactivation, 4 μl of RNA solution was used for RT-qPCR using AgPath-ID One Step kit (Life Technologies) in 10 μl reaction volume. For determination of nuclear and cytoplasmic RNA content in YTHDC1 knockdown experiments, ~2 million control LCLs were nucleoporated with YTHDC1 siRNA or Non-Targeting siRNA#1 (Silencer Select, Invitrogen) using Nucleofector II (Lonza) and Cell Line Nucleofector Kit V (Lonza) using program X005. After 48 h, RNA was isolated and subjected to RT-qPCR as above.

## Cell transfection

For plasmid DNAs, HEK293T and U87-MG cells were transfected with lipofectamine 2000 for 24–72 h. For siRNAs, HEK293T cells were transfected with lipofectamine RNAiMAX and 50–100 nM targeting siRNAs or Non-Targeting siRNA #1 (Silencer Select, Invitrogen) for 72 h. Transfection mixes were made in OptiMEM-I and added to cells in fresh growth medium.

## Luciferase assay

HEK293T and U87-MG cells were co-transfected with hRuc reporter and FLuc control plasmids using Lipofectamine 2000 for 24 h. Following cell lysis with Passive Lysis Buffer (Promega) for 20 min according to the manufacturer's protocol, *Renilla* and firefly luciferase activities were determined using Renilla Glo and Luciferase Assay Systems (Promega), respectively, using a Wallac Victor3 1420 multi-label counter system and Wallac 1420 Manager v3 (Perkin Elmer), or Spectramax i3X system and SoftMax Pro v6.5.1 (Molecular Devices).

## Immunoprecipitation of m⁶A-modified RNA (MeRIP) and RT-qPCR

Total RNA was isolated from HEK 293 T and U87-MG cells transfected with DRACH-less hRLuc plasmids, or from LCLs untreated or treated with PMOs or TRM editors. Per 30 μg total RNA to be immunoprecipitated, 1 μg of anti-m⁶A antibody (Synaptic Systems) or rabbit IgG isotype control (Cell Signaling) and 5 μl protein A/G Dynabeads (Invitrogen) were incubated in binding buffer containing 50 mM Tris pH 7.6, 50 mM NaCl, 1 mM DTT, and 100 U/ml RnaseOUT for 30 min at room temperature, followed by 30 min at 4 °C with rotation. The bead-antibody slurry was washed twice and resuspended in ice-cold binding buffer. 10–60 μg of total RNA was used for immunoprecipitation with bead-antibody slurry in a final volume of 200–400 μl of ice-cold binding buffer. Tubes were rotated at 4 °C for 2 h, followed by three washes in ice-cold binding buffer on a chilled magnetic separator. Washed beads were resuspended in Trizol, vortexed 20 s, and stored at −80 °C. Extraction of m⁶A-modified RNA was done using RNeasy Mini kit (Qiagen) with on-column Dnase-I digestion. Equal volumes of eluted RNA were used in one-step RT-qPCR with Ag-Path ID Kit. For reporter-transfected cells, fold-change in $\Delta\Delta C_t$ values was obtained for *Renilla* mRNA (refer to Supplementary Table 4 for details of probe-primer sets) with *GAPDH* mRNA as control, from anti-m⁶A-IP compared to IgG-IP. For LCLs, fold-change in $\Delta\Delta C_t$ values was obtained for *EPRS1* mRNA with *ACTB* mRNA as control, from anti-m⁶A-IP compared to IgG-IP for sibling LCLs. Similarly derived values from control LCLs were used as baseline to calculate fold-change of *EPRS1* mRNA immunoprecipitated with anti-m⁶A antibody.

## RNA Immunoprecipitation (RIP) and RT-qPCR

LCLs were harvested by centrifugation at 200 g for 5 min and washed once with PBS. Pellets were lysed in ~300 μl IP buffer (20 mM Tris pH 7.5, 100 mM KCl, 5 mM MgCl₂, 10 mM sodium orthovanadate, 0.2% Triton X-100, 1 mM DTT, 1X Halt protease and phosphatase inhibitors (Thermo), and 100 U/ml RNaseOUT) per $10^6$ cells. Cells were lysed by gently pipetting the pellet 10 times and then mixing in an end-to-end rocker at 4 °C for ~45 min. The supernatant was collected after centrifugation at $13,000 \times g$ for 5 min at 4 °C. Cell extracts were diluted to halve Triton X-100 concentration using detergent-free IP buffer and incubated with IgG control (Cell Signaling) or anti-YTHDF1/DF2/DF3/DC1/DC2 antibodies for 6 h to overnight at 4 °C, and then incubated with A/G Dynabeads for 2 h at 4 °C. The beads were washed three times in detergent-free IP buffer with RNaseOUT (100 U/ml) and then incubated with proteinase K (30 μg, Ambion) in IP buffer (detergent-free, protease- and phosphatase inhibitor-free) containing 0.1% SDS. After 30 min digestion at 55 °C, Trizol was added to the beads, and RNA isolated using RNeasy Mini kit (Qiagen) with on-column Dnase-I (Qiagen) digestion. Equal volumes of eluted RNA were used in one-step RT-qPCR with Ag-Path ID Kit. Fold-change in $\Delta\Delta C_t$ values was obtained for *EPRS1* mRNA with *ACTB* mRNA as control, from IP with YTHDF1/DF2/DF3/DC1/DC2 compared to IP with IgG for sibling LCLs. Similarly derived values from control LCLs were used as baseline to calculate fold-change of *EPRS1* mRNA co-immunoprecipitated in RNP complexes with the m⁶A reader YTH proteins.

## PMO design and *in cellulae* treatment

PMOs were generated antisense to specific RNA regions near the m⁶A and c.4444 C > A variant sites (Gene Tools, OR; refer to Supplementary Table 4 for sequences). LCLs (~2 million/well) were seeded in 6-well plates in 2 ml of growth medium. PMOs (10 μM for singleton or 5 μM each for co-treatments) were added, followed by EndoPorter-PEG delivery reagent (8 μM) according to manufacturer's protocol. After 48 h, ~1.4 ml of spent medium was carefully removed, replaced with same volume of fresh medium, and PMO delivery with EndoPorter-PEG was repeated. After 48 h, media was removed and PMO delivered a third time. Finally, after another 48 h for singleton PMO treatments (a total of

6 days from first treatment), or 72 h for PMO co-treatment (a total of 7 days), triply-dosed LCLs were harvested for further analysis.

## Nucleofection of LCLs

For knockdown experiments, ~2 million control LCLs were nucleoporated with 300 nM targeting siRNA or Non-Targeting siRNA#1 (Silencer Select, Invitrogen), using program X05 on Nucleofector I or X005 on Nucleofector II, and Cell Line Nucleofector Kit V (Lonza), and cells collected after 72–75 h. In experiments comparing single and double knockdowns, 200 nM of targeting siRNAs or Non-Targeting siRNA#1 was used in single knockdowns, whereas 150 nM of each siRNA was used for double knockdowns. For experiments with nuclear TRM editors, ~2 million LCLs were nucleofected with a mix of 95% crRNA encoding pU6-PspCas13b plasmid and pCMV-dCas13b-METTL3-NLS or dCas13b-METTL3$^{mut}$-NLS plasmid at 1:2 molarity, and 5% pmaxGFP plasmid (Lonza) at a final amount of 2 µg plasmid DNA per nucleofection, using program X05 on Nucleofector I and Nucleofector Kit V (Lonza), and cells harvested after 96 h. For experiments using the nuclear dCasRX-ALKBH5 eraser, ~2 million LCLs were nucleofected with a mix of 95% crRNA encoding pXR003 plasmid and pMSCV-dCasRx-ALKBH5-PURO plasmid at a 1:1 ratio of DNA and 5% pmaxGFP plasmid at a final amount of 2 µg plasmid DNA per nucleofection using kit and program as above, and cells harvested after 72 h.

## RNA structure prediction

Mfold RNA Folding Form v2.3, RNAstructure v6.4, and RNAfold web server (ViennaRNA Package 2.0) energy minimization algorithms[53,104,105] were used to fold RNA sequences.

## Evolutionary conservation analysis of G$_4$A:UC$_4$ stem in *EPRS1* mRNA

Gene ID for *EPRS1* from multiple species were retrieved from NCBI Orthologs. Genomic sequences were aligned by Clustal Omega v1.2.4[106]. 24-nt windows, in-frame with the coding sequence, in the orthologous region surrounding the C4444A variant site (UC$_4$) arm and the opposite strand (G$_4$A) were curated. Frequency plots were generated on WebLogo3 v2.8.2[107] and encoded amino acids annotated. Base-pair conservation in greater than 90% of species was considered strong.

| Common name (scientific name) | *EPRS1* Gene ID (NCBI) |
|---|---|
| **Placental mammals (Eutheria)** | |
| Human (*Homo sapiens*) | 2058 |
| House mouse (*Mus musculus*) | 107508 |
| Black rat (*Rattus norvegicus*) | 289352 |
| Small eared galago (*Otolemur garnettii*) | 100966256 |
| Ring-tailed lemur (*Lemur catta*) | 123627005 |
| Gray mouse lemur (*Microcebus murinus*) | 105886059 |
| Philippine tarsier (*Carlito syrichta*) | 103257617 |
| Tufted capuchin (*Sapajus apella*) | 116558954 |
| Rhesus monkey (*Macaca mulatta*) | 706899 |
| Chimpanzee (*Pan troglodytes*) | 457746 |
| Western lowland gorilla (*Gorilla gorilla gorilla*) | 101132255 |
| Big brown bat (*Eptesicus fuscus*) | 103293465 |
| Chinese tree shrew (*Tupaia chinensis*) | 102472024 |
| Sunda flying lemur (*Galeopterus variegatus*) | 103603328 |
| American beaver (*Castor canadensis*) | 109682484 |

| | |
|---|---|
| Gray squirrel (*Sciurus carolinensis*) | 124962055 |
| Rabbit (*Oryctolagus cuniculus*) | 100340173 |
| Dog (*Canis lupus familiaris*) | 478962 |
| Beluga whale (*Delphinapterus leucas*) | 111177482 |
| Common bottlenose dolphin (*Tursiops truncatus*) | 101332069 |
| Cow (*Bos taurus*) | 538357 |
| Sheep (*Ovis aries*) | 101104590 |
| Common vampire bat (*Desmodus rotundus*) | 112318748 |
| **Reptiles (Reptilia)** | |
| Tiger rattlesnake (*Crotalus tigris*) | 120307188 |
| Mainland tigersnake (*Notechis scutatus*) | 113421898 |
| Western terrestrial garter snake (*Thamnophis elegans*) | 116507382 |
| Common garter snake (*Thamnophis sirtalis*) | 106547182 |
| Chinese alligator (*Alligator sinensis*) | 102381974 |
| American alligator (*Alligator mississippiensis*) | 102565153 |
| Reeve's turtle (*Mauremys mutica*) | 120400723 |

## SELECT-qPCR (single-base elongation- and ligation-based quantitative PCR)

The protocol was adapted from ref. 55 For each interrogated DRACH site, ~1.25 µg of Trizol-extracted, DNase-treated, and column-purified total RNA from control or patient LCLs was mixed with 40 nM 5'-phosphorylated site-specific down-probe, 40 nM site-specific up-probe (refer to Supplementary Table 4 for sequences), and 5 µM dTTP in 17 µl 1xCutSmart buffer (NEB) and DEPC-treated, autoclaved water. The RNA and primer mixtures were annealed by incubating at a strand-denaturing temperature gradient on a thermal cycler as follows: 90 °C for 1 min, 80 °C for 1 min, 70 °C for 1 min, 60 °C for 1 min, 50 °C for 1 min, and then 40 °C for 6 min. Bst 2.0 DNA polymerase (NEB) and SplintR ligase (NEB) were diluted with Diluent A (NEB) to final concentrations of 0.01 U/µL and 0.5 U/µL, respectively. A 3-µl mixture containing 0.01 U DNA polymerase, 0.5 U ligase, and 10 nmol rATP was added to the mixture above to a final volume of 20 µl. The final reaction mixture was incubated at 40 °C for 20 min, denatured at 80 °C for 20 min, and held at 4 °C on a thermal cycler. qPCR reaction was done using StepOnePlus Real-Time PCR System and StepOne v2.3 (Applied Biosystems). A 10 µl qPCR reaction contained 5 µl of 2X PowerUp SYBR Green Master Mix (Applied Biosystems), 200 nM SELECTq PCR forward and reverse primers, 2 µl of the final SELECT reaction mixture, and DEPC-treated, autoclaved water. qPCR program was as follows: 50 °C, 2 min; 95 °C, 5 min; (95 °C, 10 s; 60 °C, 35 s) × 40 cycles; 95 °C, 15 s; 60 °C, 1 min; 95 °C, 15 s (fluorescence collected for melt curve on continuous mode); and 4 °C, 15 min.

For SELECT-qPCR from *EPRS1* knockdown of control LCLs, cells were nucleofected with 300 nM siRNA (Non-targeting or *EPRS1*-targeting) for 96 hr and total RNAs isolated and processed as above. For FTO-mediated demethylation, FTO was purchased from ActiveMotif and protocol was adapted from ref. 55 and instructions from manufacturer. ~30 µg of total RNA from control LCLs was incubated with 50 mM HEPES pH 7.5, 2 mM ascorbate, 300 µM α-ketoglutarate, 283 µM $(NH_4)_2Fe(SO_4)_2 \cdot 6H_2O$ (Mohr's salt), 0.2 U/µL RNaseOUT, 1.1 mM DTT, and ~0.8 µM FTO in a 50 µl reaction at 37 °C for 60–90 min. Reactions were quenched by adding 0.5 M EDTA to a final concentration of 25 mM. For pre-quenched reactions, 25 mM EDTA was included at the outset. After inactivation at 95 °C for 5 min, tubes were cooled to ambient temperature, and RNA was Trizol-extracted and processed as for SELECT-qPCR.

## Prediction of m⁶Ad-SNVs and gene ontology analysis

The pipeline described here generates a comprehensive software solution implemented in Python[108] for analysis of variant-dependent m⁶A modifications within the human genome. The software takes as input three distinct files: a FASTA file containing the human genome sequences, a VCF file containing a curated list of variations sourced from the ClinVar database[109] intersected with RMVar[110] to focus on m⁶A modifications only, and a BED file specifying a series of genomic regions of interest. Human genome version GRCh38 was used, and the BED file contains genomic coordinates of protein-coding, whole gene regions with transcripts from NCBI RefSeq extracted from the UCSC Genome Browser[111].

Initially, the BED file is processed to extract, for each listed transcript, the genomic coordinates of the last two coding sequence exons, along with up to 100 base pairs of untranslated regions (UTRs) adjacent to the CDS. Once the BED file is processed, the pipeline iterates through the VCF entries with the m⁶A modifications. The occurrence of each modification in the BED file is verified across the regions described above. Upon a positive match, the nucleotide sequences of the corresponding CDS and UTR segments, per the relevant transcript, are retrieved from the input genome FASTA file, yielding a reference sequence. Subsequently, an alternate sequence is generated by applying the modification specified in the VCF entry. The pipeline proceeds by searching for DRACH sites within the reference sequence through a regular expression (DRACH → [AGT][AG]AC[ACT]). Modifications overlapping DRACH motifs, causing termination codons or frameshifts, or altering termination codons to sense codons, are rejected. Following these quality filters, we predict the secondary structure of the reference and alternate sequences using ViennaRNA RNAfold[105], with a single constraint that isolated base pairs are not formed.

The primary objective is to assess if m⁶A-distal SNVs affect DRACH site accessibility, specifically by evaluating the alteration of base-pairing of nucleotides within segments of the DRACH motif, i.e., DRA, RAC, ACH, DRAC, RACH, and the entire DRACH. m⁶A-site accessibility, as reflected by DRACH-site base-pairing, was used as a scoring system for ranking the identified targets. The evaluation compares the reference and alternate dot-bracket representation of the folding structures at the DRACH level, with non-overlapping scoring, e.g., in the case of a free DRAC site, the counter of free DRAC sites is incremented, but the same site is not considered for counting free DRA and RAC sites. The final pipeline output is a table that reports filtered m⁶A sites with altered availability, and relevant information, including ClinVar ID, genomic coordinates, strand orientation, associated gene symbol, NCBI RefSeq transcript ID, modification position, minimum free energy (MFE) for reference and alternate structures, ΔMFE (difference between the absolute values of MFEs), a flag reporting whether the modification is synonymous, the total number of DRACH sites in both the reference and alternate structures, and the number of free and paired DRA, RAC, ACH, DRAC, RACH, and DRACH sites in both the reference and alternate structures. The table of predicted m⁶Ad-SNV candidates is interactive and permits visual inspection of predicted structures rendered with the *forna* JavaScript library[112], with the length of the reference and m⁶Ad-SNV-containing alternate sequences limited to 250 base pairs. Within this constraint, applied to maintain reliability of the results predicted by RNAfold, the sequence composition contains up to 100 base pairs from 3′UTRs, with the remaining base pairs limited to the last two exons.

A statistical over-representation test of gene ontology terms for biological processes was performed for m⁶Ad-SNV-containing genes as well as for the parent list of ClinVar genes on PANTHER version 17.0[113], using the *Homo sapiens* reference list, Fisher's exact test type, and Bonferroni correction for multiple testing. Functional annotation clustering charts for m⁶Ad-SNV-containing genes and ClinVar genes were generated from DAVID Functional Annotation Tool (DAVID 2021 with DAVID Knowlegebase v2023q4) at a threshold of five genes in any cluster[114,115], and clustering terms at a cut-off of $p < 0.04$ were included.

## Molecular dynamic simulation (MDS) of WT and P1482T ProRS homodimers

A 100-ns MDS was performed on the WT and Pro1482Thr variant form of the ProRS dimer using GROMACS v2023.1[116]. A structural model was constructed from the crystal structure of human ProRS dimer (PDB ID: 4HVC)[19] plus the addition of peptide regions absent from the crystal structure (T1312A-A1314A, D1464A-S1473A, G1498A-K1499A, T1312B-L1315B, R1463B-S1473B) using SWISS-MODEL webserver with ProMod3 modeling engine v3.4.0[117]. The model of the P1482T mutant of ProRS was generated by mutation of both dimer chains the using the molecular visualization program Pymol v2.5.0[118]. The ATP ligand was modified from phosphoaminophosphonic acid, and the proline ligand was added by overlapping the 4HVC structure with the ProRS dimer structure from PDB id: 5VAD[119]. Both molecular models were prepared for simulation at pH 7.0 by protonating lysine and arginine and deprotonating aspartate and glutamate. The models were separately inserted in a simulated box of water such that the distance from the extremity of the model to the box was 10 Å. Net protein charge (negative 26) was neutralized by random replacement of 26 water molecules with Na⁺. To simulate a physiological medium of 0.154 M NaCl, additional NaCl ion pairs randomly replaced water molecules. The model was relaxed by geometry optimization to remove steric tensions, and a series of short molecular dynamics simulations were done in which protein atoms were initially restrained permitting relaxation the water molecules and free ions. Finally, before the production simulation, a 200 ps equilibration simulation was performed to facilitate relaxation of the whole system. The production simulation was performed for 100 ns at room temperature (298 K) and 1 atm. The velocities were integrated with a time step of 2 fs and a simulation frame was saved every 10 ps, generating a trajectory of 5 GB. At the end of the simulation, frames were extracted from the trajectory, and the final frames in the simulations (100 ns) were used to generate figures. Official validation reports of both PDB structures are provided in figshare repository [https://doi.org/10.6084/m9.figshare.25607931].

## Quantification and statistical analysis

Statistical analysis was performed using GraphPad Prism 9 and 10 (GraphPad Software). Densitometric quantification was done with ImageJ v1.51k and Fiji v2.0.0-rc-69/1.52n. Replicate number is described in figure legends. Data are plotted as mean ± SD, statistical significance was calculated using Student's unpaired *t*-test, except for Figs. 2e, 3e (right panel), and Supplementary Figs. 4b, c, and 16b, where one-sample *t*-tests were appropriate. Statistical significance of key conditions is shown as *p* values. For quantitative assays, biological replicates were employed, and results were reliably replicated across at least three independent biological replicates. An outlier correction was used as noted in legend for Supplementary Fig. 7b. No quantitative experiment reported is solely from technical replicates.

## Reporting summary

Further information on research design is available in the Nature Portfolio Reporting Summary linked to this article.

# Data availability

Source data are provided in figshare repository [https://doi.org/10.6084/m9.figshare.25607931]. All graph data used in this study are available in figshare repository [https://doi.org/10.6084/m9.figshare.25607931]. All raw micrographs used in Figures and Supplementary Figs. are available in the in figshare repository [https://doi.org/10.6084/m9.figshare.25607931]. All oligonucleotide sequences (PMOs, primers, SELECT-qPCR probes) and plasmid-based CRISPR gRNA spacer sequences are detailed in Supplementary Table 4 in

Supplementary Information file. The ClinVar ID for the *EPRS1* variant is 3069175 (accession number: SCV004809027) [https://www.ncbi.nlm.nih.gov/clinvar/variation/3069175/?oq=SCV004809027&m=NM_004446.3(EPRS1):c.4444C%3EA%20(p.Pro1482Thr)]. *EPRS1* gene IDs used for alignment in Supplementary Fig. 10b are provided in Methods section as a table. Rabbit beta globin intron 2 sequence was from HBB2 [https://www.ncbi.nlm.nih.gov/gene/100009084] and chimeric intron was from pCI vector [https://www.ncbi.nlm.nih.gov/nuccore/U47119.2]. EPRS1 PDB structures used for MDS are 4HVC and 5VAD. The m⁶A RNA methylation data used in m⁶Ad-SNVs prediction is available in RMVar database [https://rmvar.renlab.org/] and used as detailed in Methods. The ClinVar variants data used in m⁶Ad-SNVs prediction is available in ClinVar database [https://www.ncbi.nlm.nih.gov/clinvar/] and used as detailed in Methods. Human reference sequence transcripts used in m⁶Ad-SNVs prediction are available in [https://www.ncbi.nlm.nih.gov/refseq/] and used as detailed in Methods. Detailed superset of Supplementary Table 3 is available at [https://doi.org/10.5281/zenodo.10905850] (interactive and machine-readable results produced by the m⁶Ad-SNVs prediction tool, contains transcript IDs, gene symbols, ClinVar ID, predicted RNA structures, and available DRACH sites). The exome sequencing was carried out in a commercial laboratory (GeneDx) for clinical purposes prior to patient enrollment in the current research study. The sequencing was performed for the purpose of clinical diagnosis and care, and consent was specifically obtained for further studies on a research basis to investigate the pathogenicity of the *EPRS1* variant. The patient exome sequencing data can be made available upon request to qualified investigators upon receipt of a written protocol which clearly states the reason for requesting access to the family's clinical exome sequencing data, proof of IRB approval of the protocol from the investigator's institution, and approval of the protocol from the SickKids Research Ethics Board, subject to both the family's consent and agreement by GeneDx to release the exome data. The contact for additional patient information is Grace Yoon (grace.yoon@utoronto.ca).

## Code availability

The code is available at [https://doi.org/10.5281/zenodo.10905411] linked to GitHub at [https://github.com/cumbof/m6Ad-SNVs] under an MIT license.

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

## Acknowledgements

We thank Dr. Charis Eng and Jennifer DeVecchio (Genomic Medicine Institute, Lerner Research Institute) for sharing control lymphoblastoid cell lines. We are grateful to all Fox laboratory members and the Cleveland Clinic Innovations team for helpful comments and suggestions. pCMV-dCas13-M3nls, pCMV-dCas13-inactive M3nls, and pU6-PspCas13b-gRNA-Actb1216 were gifts from Dr. David Liu, pMSCV-dCasRx-ALKBH5-PURO was a gift from Dr. Qi Xie, and pXR003: CasRx gRNA cloning backbone was a gift from Dr. Patrick Hsu (all obtained

from Addgene). This work was supported by NIH grants R01 DK124203 (P.L.F.), R01 DK130377 (P.L.F.), R01 AG067146 (P.L.F.), R01 NS124547 (P.L.F.), R01 NS124581 (P.L.F.) and American Heart Association Grant #19POST34380687/DebjitKhan/2019 (D.K.).

## Author contributions

P.L.F. supervised the project. G.C., S.B., A.C., and G.Y. collected and analyzed all patient data. D.K., K.V., A.C. I.R., K.K., R.D., D.H., I.Z. and B.L. designed and performed the biochemical and cell-based experiments. D.K. performed all m6A modification-related experiments. V.G. performed and analyzed the molecular dynamic simulations. D.B. and F.C. were responsible for the global, bioinformatic analysis of m6A modification. D.K., R.D., F.T., and P.L.F. were responsible for analysis of non-patient data. D.K., G.Y., F.T., F.C., D.B., and P.L.F. wrote the manuscript with help from all authors.

## Competing interests

The authors declare no competing interests.

## Additional information

[1]Department of Cardiovascular and Metabolic Sciences, Cleveland Clinic, Lerner Research Institute, Cleveland, OH, USA. [2]Genomic Medicine Institute, Cleveland Clinic, Lerner Research Institute, Cleveland, OH, USA. [3]Department of Chemistry, Cleveland State University, Cleveland, OH, USA. [4]Department of Paediatrics, Division of Clinical and Metabolic Genetics, The Hospital for Sick Children, University of Toronto, Toronto, ON, Canada. [5]Department of Diagnostic Imaging, Division of Neuroradiology, The Hospital for Sick Children, University of Toronto, Toronto, ON, Canada. [6]Department of Neuroscience, Cleveland Clinic, Lerner Research Institute, Cleveland, OH, USA. [7]Department of Paediatrics, Division of Neurology, The Hospital for Sick Children, University of Toronto, Toronto, ON, Canada. ✉e-mail: grace.yoon@utoronto.ca; foxp@ccf.org

