## [Peer Review File · Nature Communications]

Homozygous EPRS1 missense variant causing hypomyelinating leukodystrophy-15 alters variant-distal mRNA m6A site accessibilityReviewer #1 (Remarks to the Author):

In this work, the authors have identified a homozygous missense SNV in EPRS1, a cytosolic aminoacyl-tRNA synthetase (aaRSs). Previous work on this and other aaRSs have identified pathogenic variants in this class of enzymes that have been associated with spectrum of neurological diseases. In most cases, diminished tRNA charging activity is caused by mutations at or near catalytic sites, but in this work the authors present a new variant that is more distal. Based on a comprehensive set of experiments aimed at understanding the consequences of this SNV, the authors suggest that this change alters something related to the posttranscriptional regulation of EPRS1 expression. mRNA expression levels, splicing, polyadenylation, and global protein synthesis all appear similar in cells expressing wild type and the patient variant EPRS1. However reduced polysome associated EPRS1 and lower EPRS1 protein expression were observed. Given the location of the variant in the terminal exon, the authors explored the possibility that the m6A modification might be involved. Knockdown of the m6A writer METTL3 and reader YTHDC1 result in reduced EPRS1 protein expression, with the latter also resulting in increased nuclear mRNA retention. Consistent with a role for m6A in the altered expression of EPRS1 in the siblings with the SNV, they show reduced EPRS1 in m6A-antibody RNA pulldowns, as well as in YTHDC1, YTHDF1 and YTHDF3 pulldowns. However, given that the variant is not in an m6A consensus sequence and is distal from m6A sites previously annotated in online databases and based on mutation analysis, the authors propose a mechanism of action in which structural changes in the RNA alter the accessibility of m6A methylation sites. Intriguingly, the authors can also demonstrate that targeted anti-sense oligonucleotides can restore accessibility and m6A methylation. Taken together, this study presents an fascinating example where distal sequence changes can alter m6A-mediated regulation of gene expression, as well as a strategy by which this could be reversed for patient benefit. The work is thorough, technically sound, and well executed, and should be of broad interest in RNA biology and RNA therapeutics fields. I very much appreciate the detailed, stepwise approach taken to narrow down the step(s) in gene expression regulation that this SNV alters. I am thoroughly intrigued by the work and look forward to seeing it published, but I do have two points that I feel should be addressed prior to publication to clarify the mechanism and model that the authors present.

1. The authors reliance on available m6A data repositories to narrow down possible sites is not optimal. Even within the referenced atlas there are multiple annotations in the same cell line, and there are multiple such repositories that are not always consistent across one another. As the authors point out, changes in accessibility can be driven by more distal sequence elements. In addition, the experiments narrowing down the relevant m6A sites are a bit indirect. The authors should map m6A-methylated sites in the relevant cells they are using (ideally the patient-derived LCLs) to confirm changes in methylation between controls, parents, and affected siblings. This could be done with a similar approach to the m6A-IP approach used by the authors in the manuscript, or with newer higher resolution m6A sequencing approaches (miCLIP, GLORI, eTAM-seq, etc).

2. The model that the authors suggest based on their data invokes the roles of both YTHDC1, a nuclear m6A reader, and YTHDF1/YTHDF3, both cytoplasmic m6A readers. They postulate that YTHDC1 regulates

m6A-mediated export of EPRS1 and that YTHDF1/3 may regulate its translation, but the experiments related to these cytoplasmic readers are limited to just the knockdown experiments with apparently limited knockdown efficiency. The authors suggest that the export-mediated effect is not sufficient to explain the change in expression, but this has not been quantitatively or thoroughly tested. This could be determined by double knockdown experiments to determine whether combined knockdown of YTHDC1 with YTHDF1/3 has an additive effect. Another way to test this could be to see if reduced polysome associated-EPRS1 is observed with loss of YTHDC1 alone.

3. A minor suggested text change: in the section “Role of EPRS1 mRNA m6A modification in reduced expression of variant EPRS1” the phrase “methylation of the N6-position of adenosine (m6A, N6-methyladenosine), the most abundant mRNA modification...” should be changed to “the most abundant internal mRNA modification...”

Reviewer #2 (Remarks to the Author):

The manuscript by Khan et al. identified a missense mutation (c.4444C>A; p.Pro1482Thr) in EPRS1 gene from siblings with hypomyelinating leukodystrophy (HLD). Although not definite, enough evidence has been presented to suggest that the mutation is responsible for the disease. The authors found that the mutation is associated with a severe reduction in EPRS1 at the protein, but not at the mRNA, level. Defect in protein stability and enzymatic activity has been ruled out, thus suggesting a defect in EPRS1 mRNAs unable to be efficiently processed into EPRS1 proteins. Indeed, the authors found impaired nuclear export and impaired translation of the mutant mRNA, and both impairments are linked to severely reduced m6A methylation of the EPRS1 mRNAs. Interestingly, the mutation site does not overlap with the m6A methylation sites, and through comprehensive mutagenesis studies, the authors have thoroughly demonstrated that mutation causes conformational change of the mRNA, which blocks the accessibility for modification at a distal site. Furthermore, the authors deployed two different methods (i.e., antisense morpholine and METTL3-dCas13b-targeted and forced m6A methylation) and both methods were able to successfully rescue the EPRS1 expression defect in patient-derived lymphoblastoid cell lines. The success not only confirmed the proposed pathological mechanism, but also suggested potential therapeutics for the disease. The thoroughness and depth of the work are impressive. The authors went above and beyond of what is necessary for the story. The work has broad significance in considering a disease-causing missense mutation and emphasizes the importance of not only considering the impact of mutation on the protein but also on the mRNA.

Minor points:

1. Pg 12. First line – “YTHDC2 and YTHDF2 are cytoplasmic m6A readers...” this is inconsistent with Fig. 4D, where YTHDC2 is indicated a nuclear reader.
2. Based on Fig. 4B and C, C4444A does not affect methylation at site16726. Why forced m6A methylation at the 16726 site is effective in restoring EPRS1 expression (Fig. 6E)?

3. Can the authors provide some explanation for why the presence of the I31 intron synergize with the effect of the C4444A mutation? As indicated by the authors, inclusion of I31 intron does not seem to impact the folding of the mRNA.

Reviewer #3 (Remarks to the Author):

The study identified a novel c.4444C>A; p.Pro1482Thr missense single nucleotide variant in the cytoplasmic glutamyl-prolyl-tRNA synthetase gene that when homozygous causes a severe form of hypomyelinating leukodystrophy (HLD15). The study provides a very detailed characterisation of the impacts of this mutation on EPRS1 expression levels and activity. In patient-derived lymphoblastoid cells, EPRS1 protein expression was significantly decreased due to nuclear export defects and reduced translation of variant EPRS1 mRNA, both caused by hindered m6A modification of mRNA target sites. The EPRS1c.4444C>A variant does not change the sequence of m6A sites directly, but instead causes structural mRNA changes that reduce their accessibility. This represents a novel RNA-dependent mechanism by which single nucleotide variants can influence gene expression. The study further employed antisense morpholinos to re-expose EPRS1 mRNA m6A sites as well as a targeted RNA methylation editor (METTL3-dCas13b), both of which successfully restored mRNA methylation and rescued EPRS1 expression. Both approaches could potentially be used as novel therapeutics for the treatment of this and other genetic disorders caused by variants that alter accessibility of distal mRNA m6A sites. Lastly, the study conducted a bioinformatic analysis demonstrating widespread occurrence of single nucleotide variants that alter m6A accessibility indicating that this newly discovered disease mechanism could be involved in many other diseases.

This is a very comprehensive and thorough study not only investigating the consequences of the newly discovered EPRS1 c.4444C>A; p.Pro1482Thr missense single nucleotide variant but also elucidating a novel mechanism by which synonymous and non-synonymous variants can cause structural mRNA changes that reduce the accessibility and modification of mRNA m6A sites, thus resulting in impaired nuclear mRNA export and reduced translation. The study design, research, and manuscript are very well thought-through, sound, and nicely presented, and will be a great addition to Nature Communications. Upon minor revisions addressing the points raised below I recommend publication of this manuscript.

- Hypomyelinating leukodystrophies represent a group of neurogenetic disorders characterised by hypomyelination of the CNS, rather than one specific disease. EPRS1 mutations are usually associated with HLD15. Please specify in abstract and title which form of hypomyelinating leukodystrophy is caused by this EPRS1 variant.

- ProRS activity in P1482T LCLs was reduced to 20% of wildtype activity, while PheRS activity was unaffected. As EPRS1 encodes the glutamyl-prolyl-tRNA synthetase, it would be interesting to see if GluRS charging activity is altered to the same extent as ProRS activity in mutant LCLs? Considering that EPRS1 protein levels were reduced by 80% in mutant LCL lysates and that GluRS and ProRS are covalently linked, GluRS-mediated charging is likely also reduced and thus could contribute to the disease.

- Figure legend of Fig. 2c states HEK293F cells, while the main text states HEK293T cells for Fig. 2c. Please clarify.
- N numbers are missing from some figure legends. Please make sure that n numbers for all experiment are included in the figure legends.
- Include densitometry quantification of immunoblot results (Fig.3d-g) across biological replicates.
- Which statistical test was used in Fig.3?
- P-values are missing from Fig.4 b and c. Please add.
- Which statistical test was used in Fig.4?
- Fig. 5c: Include p-values.
- Probably beyond the scope of the study but it will be intriguing to see whether the antisense morpholinos exposing the m6A sites or METTL3-dCas13b-mediated modification of m6A can rescue functional deficits in an EPRS1P1482T animal model (e.g., mouse) or in brain organoids from patient-derived iPSCs. Likewise, follow-up studies investigating the downstream functional consequences of reduced EPRS1 levels and activity will be essential to understand the pathophysiology of this form of aminoacyl-tRNA synthetase deficiency and help to understand why the CNS is more profoundly affected than other organs.

We are grateful to the three Reviewers for their positive responses to the work overall and for their insightful and helpful comments that have improved the clarity and rigor of the manuscript. The Reviewer's comments are shown in full in italics, and our responses in plain font:

REVIEWER COMMENTS

Reviewer #1 (Remarks to the Author):

In this work, the authors have identified a homozygous missense SNV in EPRS1, a cytosolic aminoacyl-tRNA synthetase (aaRSs). Previous work on this and other aaRSs have identified pathogenic variants in this class of enzymes that have been associated with spectrum of neurological diseases. In most cases, diminished tRNA charging activity is caused by mutations at or near catalytic sites, but in this work the authors present a new variant that is more distal. Based on a comprehensive set of experiments aimed at understanding the consequences of this SNV, the authors suggest that this change alters something related to the posttranscriptional regulation of EPRS1 expression. mRNA expression levels, splicing, polyadenylation, and global protein synthesis all appear similar in cells expressing wild type and the patient variant EPRS1. However reduced polysome associated EPRS1 and lower EPRS1 protein expression were observed. Given the location of the variant in the terminal exon, the authors explored the possibility that the m6A modification might be involved. Knockdown of the m6A writer METTL3 and reader YTHDC1 result in reduced EPRS1 protein expression, with the latter also resulting in increased nuclear mRNA retention. Consistent with a role for m6A in the altered expression of EPRS1 in the siblings with the SNV, they show reduced EPRS1 in m6A-antibody RNA pull-downs, as well as in YTHDC1, YTHDF1 and YTHDF3 pull-downs. However, given that the variant is not in an m6A consensus sequence and is distal from m6A sites previously annotated in online databases and based on mutation analysis, the authors propose a mechanism of action in which structural changes in the RNA alter the accessibility of m6A methylation sites. Intriguingly, the authors can also demonstrate that targeted anti-sense oligonucleotides can restore accessibility and m6A methylation. Taken together, this study presents an fascinating example where distal sequence changes can alter m6A-mediated regulation of gene expression, as well as a strategy by which this could be reversed for patient benefit. The work is thorough, technically sound, and well executed, and should be of broad interest in RNA biology and RNA therapeutics fields. I very much appreciate the detailed, stepwise approach taken to narrow down the step(s) in gene expression regulation that this SNV alters. I am thoroughly intrigued by the work and look forward to seeing it published, but I do have two points that I feel should be addressed prior to publication to clarify the mechanism and model that the authors present.

Response: We thank the Reviewer for the extremely positive and thoughtful summary and comments.

1. The authors reliance on available m6A data repositories to narrow down possible sites is not optimal. Even within the referenced atlas there are multiple annotations in the same cell line, and there are multiple such repositories that are not always consistent across one another. As the authors point out, changes in accessibility can be driven by more distal sequence elements. In addition, the experiments narrowing down the relevant m6A sites are a bit indirect. The authors should map m6A-methylated sites in the relevant cells they are using (ideally the patient-derived LCLs) to confirm changes in methylation between controls, parents, and affected siblings. This could be done with a similar approach to the m6A-IP approach used by the authors in the manuscript, or with newer higher resolution m6A sequencing approaches (miCLIP, GLORI, eTAM-seq, etc).

Response: We thank the Reviewer for this important observation. Indeed, the last two exons in *EPRS1* mRNA contain thirteen DRACH sequences (in bold in Supp. Fig. 11a), and three potential polymethylated m⁶A regions (colored boxes in Supp. Fig. 11a) where more than one DRACH sequence is present in a window of ~20-25 nts, each including site 16728, 16727, or 16726.

As suggested by the Reviewer, we interrogated specific differences in m⁶A methylation in patient LCLs compared to control LCLs in thirteen DRACH sequences in the region of interest. As an alternative to sequencing, we have used SELECT-qPCR (single-base elongation- and ligation-based qPCR) (¹, highlighted in ²), extensively used in the literature, with the necessary rigor to show signals are specific for both target mRNA and m⁶A modification. In the SELECT method, the extension and ligation steps, that covalently link two synthetic antisense probes directly flanking an adenosine residue, are hindered if the residue is N⁶-methylated, resulting in lower amounts of linked template from methylated substrates ¹. In the subsequent qPCR step, quantitative amplification with primers complementary to invariant terminal sequences of the ligated probe-pair template, identical for all tested sites, reveals changes in methylation status of the target adenosine. To improve rigor, we performed SELECT-qPCR (a) in the presence of *EPRS1* knockdown to test specificity of SELECT probes towards *EPRS1* mRNA (Supplementary Fig. 12), and (b) in presence of FTO demethylase (m⁶A-eraser) treatment of total RNA to test specificity towards m⁶A-methylation at relevant interrogated sites (Supplementary Fig. 13) per established protocols ¹.

SELECT-qPCR revealed multiple hypomethylated DRACH sites in the terminal exons of *EPRS1* mRNA in patient LCLs (Fig. 5d and Supplementary Fig. 11b). Specifically, in patient LCLs, methylation at the sites addressed in the original manuscript, i.e., 16728 (A4355) and 16727 (A4464) were lower by ~58% and ~29%, respectively (Fig. 5d, 1st and 2nd panels), while methylation at site 16726 (A4690) was nearly unaltered (Fig. 5d, 3rd panel). m⁶A methylation at a validated site on 28S rRNA ¹ was unaffected in patients LCLs (Fig. 5d, 4th panel), showing that m⁶A methylation was not altered globally. Furthermore, SELECT-qPCR targeted at control, non-DRACH adenosine residues near sites 16728 and 16726, i.e., A4349 and A4469, respectively, ruled out altered probe-pair accessibility in denatured template RNAs at either site (Fig. 5d, 5th and 6th panels). We verified that *EPRS1* mRNA levels, as well as *GAPDH* and *ACTB* mRNAs, were similar in inputs of control and patient LCLs (Fig. 5d, 7th panel and Supplementary Fig. 11b, last two panels). Unexpectedly, SELECT-qPCR revealed significant hypomethylation at A4704 and A4716 near the 3'UTR terminus of *EPRS1*, by an amount of ~40% (Supplementary Fig. 11b). We tested the requirement of these sites for protein expression in the WT C4444 reporter system used earlier to test sites 16728 and 16727 (Fig. 4a-b). As before, simultaneous disruption of the 16728 and 16727 sites (A4355/A4464) abrogated luciferase expression, but mutation of the A4704/A4716 pair did not alter expression (Supp. Fig. 15), suggesting that alterations in local RNA structure at sites 16728 and 16727 are sufficient to explain C4444A-related changes in *EPRS1* expression. Hypomethylation at A4614 and A4666 in patient LCLs was not significant, as there was a large variation within the control LCL samples (Supplementary Fig. 11b). Lastly, A4404 was hypermethylated by ~45% in patient cells. We failed to detect a specific SELECT-qPCR signal at the terminal (13th site) in *EPRS1* mRNA, possibly owing to its proximity to the poly-A site that requires a low specificity poly-T up-probe, resulting in a multi-species melt-curve upon qPCR.

To validate that probe-pairs for multiple relevant sites were specific to *EPRS1* mRNA, *EPRS1* was knocked down in control LCLs (Supplementary Fig. 12a) and SELECT-qPCR was performed on total RNA isolated from *EPRS1* knockdown cells (Supplementary Fig. 12b). An increase in C_t compared to total RNA isolated from control LCLs nucleofected with non-targeting siRNA suggested that the probe-pairs were specific, except for site A4614 in one control LCL

that confirms the variation of SELECT-qPCR signal at this site seen earlier (Supplementary Fig. 11b). Further, to validate that the signals were specific for m⁶A methylation at relevant adenosine residues, SELECT-qPCR was performed from FTO demethylase-treated total RNA from control LCLs and compared to EDTA-pre-quenched, enzyme-inactivated sets (Supplementary Fig. 13a-b). FTO treatment decreased C_t, validating m⁶A methylation-specific SELECT-qPCR signals at the relevant sites tested for EPRS1, and the 28S rRNA methylation control, except for the control site A4349 (non-DRACH), as expected (Supplementary Fig. 13b).

The SELECT results show hypomethylation at sites 16728 and 16727, but not site 16726, in patient LCLs corroborating results from reporter assays and MeRIP-RT-qPCR (Fig. 4b-c and Supplementary Fig. 9a-b). The absence of an effect of A4704 and A4716 hypomethylation on *EPRS1* reporter expression sites suggests that modification of sites 16728 and 16727 are sufficient to account for altered *EPRS1* expression. To further probe the sufficiency of these sites, we explored the effect of site-specific demethylation. Guide RNAs targeting these sites singly or in combination were nucleofected into control LCLs with a nucleus-targeted, catalytically dead RfxCas13d (dCasRx)-ALKBH5 demethylase fusion protein³⁻⁵ (Supplementary Fig. 16a). Guiding the demethylase to either site 16728 or 16727 did not reduce *EPRS1* expression, but when targeted together decreased *EPRS1* levels by about 50% - reaching 70-80% in several replicates, close to the ~75% decrease in *EPRS1* expression in sibling LCLs, and further supporting the sufficiency of these sites to account for observed loss of *EPRS1* expression in patient cells (Supplementary Fig. 16b).

2. The model that the authors suggest based on their data invokes the roles of both YTHDC1, a nuclear m⁶A reader, and YTHDF1/YTHDF3, both cytoplasmic m⁶A readers. They postulate that YTHDC1 regulates m⁶A-mediated export of EPRS1 and that YTHDF1/3 may regulate its translation, but the experiments related to these cytoplasmic readers are limited to just the knockdown experiments with apparently limited knockdown efficiency. The authors suggest that the export-mediated effect is not sufficient to explain the change in expression, but this has not been quantitatively or thoroughly tested. This could be determined by double knockdown experiments to determine whether combined knockdown of YTHDC1 with YTHDF1/3 has an additive effect. Another way to test this could be to see if reduced polysome associated-EPRS1 is observed with loss of YTHDC1 alone.

Response: As suggested, to address relative and additive roles of nuclear and cytoplasmic m⁶A readers as determinants of *EPRS1* expression, double knockdown experiments were done in control LCLs. Individual knockdowns of YTHDC1 and YTHDF1 decreased *EPRS1* expression by ~70%; added in combination expression was additively reduced by nearly 90% (Supplementary Fig. 7a). Likewise, individual knockdown of YTHDC1 and YTHDF3 reduced *EPRS1* expression comparably, but double knockdown compounded the inhibition (Supplementary Fig. 7b). Along with Fig. 4f-h (and Supplementary Figs. 4-6), this new evidence suggests that optimal *EPRS1* expression in healthy cells requires both YTHDC1-mediated nuclear export via NXF1, followed by YTHDF1/3-assisted translation.

3. A minor suggested text change: in the section "Role of EPRS1 mRNA m⁶A modification in reduced expression of variant EPRS1" the phrase "methylation of the N6-position of adenosine (m⁶A, N6-methyladenosine), the most abundant mRNA modification..." should be changed to "the most abundant internal mRNA modification..."

Response: We have edited this sentence now as suggested.

Reviewer #2 (Remarks to the Author):

The manuscript by Khan et al. identified a missense mutation (c.4444C>A; p.Pro1482Thr) in EPRS1 gene from siblings with hypomyelinating leukodystrophy (HLD). Although not definite, enough evidence has been presented to suggest that the mutation is responsible for the disease. The authors found that the mutation is associated with a severe reduction in EPRS1 at the protein, but not at the mRNA, level. Defect in protein stability and enzymatic activity has been ruled out, thus suggesting a defect in EPRS1 mRNAs unable to be efficiently processed into EPRS1 proteins. Indeed, the authors found impaired nuclear export and impaired translation of the mutant mRNA, and both impairments are linked to severely reduced m6A methylation of the EPRS1 mRNAs. Interestingly, the mutation site does not overlap with the m6A methylation sites, and through comprehensive mutagenesis studies, the authors have thoroughly demonstrated that mutation causes conformational change of the mRNA, which blocks the accessibility for modification at a distal site. Furthermore, the authors deployed two different methods (i.e., antisense morpholine and METTL3-dCas13b-targeted and forced m6A methylation) and both methods were able to successfully rescue the EPRS1 expression defect in patient-derived lymphoblastoid cell lines. The success not only confirmed the proposed pathological mechanism, but also suggested potential therapeutics for the disease. The thoroughness and depth of the work are impressive. The authors went above and beyond of what is necessary for the story. The work has broad significance in considering a disease-causing missense mutation and emphasizes the importance of not only considering the impact of mutation on the protein but also on the mRNA.

Response: We thank the Reviewer for the strong praise for our work.

Minor points:

1. Pg 12. First line – “YTHDC2 and YTHDF2 are cytoplasmic m6A readers...” this is inconsistent with Fig. 4D, where YTHDC2 is indicated a nuclear reader.

Response: We have corrected this sentence.

2. Based on Fig. 4B and C, C4444A does not affect methylation at site 16726. Why forced m6A methylation at the 16726 site is effective in restoring EPRS1 expression (Fig. 6E)?

Response: The reviewer raises an interesting point. Possibly, forced m⁶A modification at the 16726 site in the 3'UTR induces binding of m⁶A readers at sites distinct from the CDS sites near 16727/8. Also, as the site is within a polymethylation region (Supplementary Fig. 11a), site 16726 methylation might increase availability and modification of the nearby sites, further inducing binding of readers that enhance EPRS1 expression. Although intriguing, we have not pursued the 16726 site in detail since our experiments show its modification state is not altered by C4444A mutation in the siblings. Nonetheless, forced modification of this site remains a viable therapeutic target, and is now briefly described in the Discussion.

3. Can the authors provide some explanation for why the presence of the I31 intron synergize with the effect of the C4444A mutation? As indicated by the authors, inclusion of I31 intron does not seem to impact the folding of the mRNA.

Response: The Reviewer raises an important question that is incompletely understood at this time. We show that maximal inhibition of reporter expression by C4444A mutation requires an intron between exons 31 and 32, but the sequence of the intron is not critical, highlighting the

specific role of the terminal exon-exon junction (Fig. 3b, Supplementary Fig. 3b). Possibly, the intron guides specificity of m⁶A methylation at physiologically relevant sites in the gene-end architecture, spatially regulated by deposition of exon-junction complexes⁶. Interplay of nuclear reader YTHDC1 with splice adaptors and the mRNA export pathway has been reported⁷. While the mechanistic details are indeed important, and a critical missing concept for the field in general, we suggest that elucidation is beyond the scope of our investigation. Nonetheless, a brief summary of these points has been added to the Results.

Reviewer #3 (Remarks to the Author):

The study identified a novel c.4444C>A; p.Pro1482Thr missense single nucleotide variant in the cytoplasmic glutamyl-prolyl-tRNA synthetase gene that when homozygous causes a severe form of hypomyelinating leukodystrophy (HLD15). The study provides a very detailed characterisation of the impacts of this mutation on EPRS1 expression levels and activity. In patient-derived lymphoblastoid cells, EPRS1 protein expression was significantly decreased due to nuclear export defects and reduced translation of variant EPRS1 mRNA, both caused by hindered m6A modification of mRNA target sites. The EPRS1^{c.4444C>A} variant does not change the sequence of m6A sites directly, but instead causes structural mRNA changes that reduce their accessibility. This represents a novel RNA-dependent mechanism by which single nucleotide variants can influence gene expression. The study further employed antisense morpholinos to re-expose EPRS1 mRNA m6A sites as well as a targeted RNA methylation editor (METTL3-dCas13b), both of which successfully restored mRNA methylation and rescued EPRS1 expression. Both approaches could potentially be used as novel therapeutics for the treatment of this and other genetic disorders caused by variants that alter accessibility of distal mRNA m6A sites. Lastly, the study conducted a bioinformatic analysis demonstrating widespread occurrence of single nucleotide variants that alter m6A accessibility indicating that this newly discovered disease mechanism could be involved in many other diseases. This is a very comprehensive and thorough study not only investigating the consequences of the newly discovered EPRS1 c.4444C>A; p.Pro1482Thr missense single nucleotide variant but also elucidating a novel mechanism by which synonymous and non-synonymous variants can cause structural mRNA changes that reduce the accessibility and modification of mRNA m6A sites, thus resulting in impaired nuclear mRNA export and reduced translation. The study design, research, and manuscript are very well thought-through, sound, and nicely presented, and will be a great addition to Nature Communications. Upon minor revisions addressing the points raised below I recommend publication of this manuscript.

Response: We are grateful to the Reviewer for the generous comments and praise.

- Hypomyelinating leukodystrophies represent a group of neurogenetic disorders characterised by hypomyelination of the CNS, rather than one specific disease. EPRS1 mutations are usually associated with HLD15. Please specify in abstract and title which form of hypomyelinating leukodystrophy is caused by this EPRS1 variant.

Response: We now specify in the abstract, title, and introduction that EPRS1 mutations are associated with HLD15.

- ProRS activity in P1482T LCLs was reduced to 20% of wildtype activity, while PheRS activity was unaffected. As EPRS1 encodes the glutamyl-prolyl-tRNA synthetase, it would be interesting to see if GluRS charging activity is altered to the same extent as ProRS activity in mutant LCLs? Considering that EPRS1 protein levels were reduced by 80% in mutant LCL lysates and that GluRS and ProRS are covalently linked, GluRS-mediated charging is likely also reduced and thus could contribute to the disease.

Response: We have now measured GluRS charging activity of patient LCL lysates in new Figure 2b (middle panel). Consistent with reduced level of EPRS1 (Fig. 2d) and ProRS charging activity in patient LCLs (Fig. 2b, left panel), GluRS charging activity is also reduced significantly in patient LCLs.

- *Figure legend of Fig. 2c states HEK293F cells, while the main text states HEK293T cells for Fig. 2c. Please clarify.*

Response: We apologize for the confusion, recombinant EPRS1 was purified in HEK293F cells as mentioned in figure legend and Methods. We have corrected main text.

- *N numbers are missing from some figure legends. Please make sure that n numbers for all experiment are included in the figure legends.*

Response: We have added n numbers in all figure legends.

- *Include densitometry quantification of immunoblot results (Fig.3d-g) across biological replicates.*

Response: We have now included densitometric quantification of immunoblots, including biological replicates, for Fig. 3d-e in new Supplementary Fig. 4, for Fig. 3f in new Supplementary Fig. 5, and for new Fig. 3g in new Supplementary Fig. 6.

- *Which statistical test was used in Fig.3?*

Response: Statistical significance in Fig. 3 was calculated using Student's *t*-test as noted in general in Methods sub-section Quantification and Statistical Analysis. We now describe tests used in all Figure legends.

- *P-values are missing from Fig.4 b and c. Please add.*

Response: Changes in luciferase expression in new Fig. 4b now represents fold-change calculated from two experiments done in parallel in 293T and U87-MG cells in six biological replicates in each cell line (shown in new Supplementary Fig. 9a). MeRIP-RT-qPCR in new Fig. 4c represents fold-change calculated from three experiments reported in new Supplementary Fig. 9b done in 293T and U87-MG cells, in two biological replicates each. Two were reported previously and one is newly added. P-values are added as suggested.

- *Which statistical test was used in Fig.4?*

Response: We now mention tests used in all Figure legends. The statistical significance in Fig. 4 was calculated using Student's unpaired *t*-test as described in the Methods sub-section on Quantification and Statistical Analysis.

- *Fig. 5c: Include p-values.*

Response: We have added p-values for Fig. 5c.

- *Probably beyond the scope of the study but it will be intriguing to see whether the antisense morpholinos exposing the m6A sites or METTL3-dCas13b-mediated modification of m6A can rescue functional deficits in an EPRS1^{P1482T} animal model (e.g., mouse) or in brain organoids*

from patient-derived iPSCs. Likewise, follow-up studies investigating the downstream functional consequences of reduced EPRS1 levels and activity will be essential to understand the pathophysiology of this form of aminoacyl-tRNA synthetase deficiency and help to understand why the CNS is more profoundly affected than other organs.

Response: We concur that these are two high-priority questions that need to be addressed, namely, (1) can the potential therapeutics we developed be used in an *in vivo* model?, and (2) what are the functional downstream consequences of reduced EPRS1? To address both questions we have successfully applied Crispr/Cas9 to generate a knock-in mouse model of EPRS1^{C4444A}. Preliminary studies indicate there is a behavioral phenotype and we propose to use the model in future treatment studies. Ongoing single-nucleus RNA seq studies on tissue from knock-in mice might provide clues to address the second question on downstream functional consequences. We concur with the Reviewer that these important studies, at a preliminary stage at this time, are beyond the scope of the current investigation which provides a detailed mechanistic elucidation of the influence of the patient C4444A mutation on EPRS1 expression.

References

1. Xiao, Y., Wang, Y., Tang, Q., Wei, L., Zhang, X. & Jia, G. An elongation- and ligation-based qPCR amplification method for the radiolabeling-free detection of locus-specific N⁶-methyladenosine modification. *Angew. Chem. Int. Ed. Engl.*, **2018**, 57, 15995-16000.
2. Boulias, K. & Greer, E. L. Biological roles of adenine methylation in RNA. *Nat. Rev. Genet.*, **2023**, 24, 143-160. PMC9974562.
3. Xia, Z., Tang, M., Ma, J. *et al.* Epitranscriptomic editing of the RNA N6-methyladenosine modification by dCasRx conjugated methyltransferase and demethylase. *Nucleic Acids Res.*, **2021**, 49, 7361-7374. PMC8287920.
4. Konermann, S., Loffy, P., Brideau, N. J., Oki, J., Shokhirev, M. N. & Hsu, P. D. Transcriptome engineering with RNA-targeting Type VI-D CRISPR Effectors. *Cell*, **2018**, 173, 665-676 e614. PMC5910255.
5. Liu, Y., Jing, P., Zhou, Y., Zhang, J., Shi, J., Zhang, M., Yang, H. & Fei, J. The effects of length and sequence of gRNA on Cas13b and Cas13d activity in vitro and in vivo. *Biotechnol J*, **2023**, 18, e2300002.
6. Uzonyi, A., Dierks, D., Nir, R. *et al.* Exclusion of m⁶A from splice-site proximal regions by the exon junction complex dictates m⁶A topologies and mRNA stability. *Mol. Cell*, **2023**, 83, 237-251 e237.
7. Roundtree, I. A., Luo, G. Z., Zhang, Z. *et al.* YTHDC1 mediates nuclear export of N⁶-methyladenosine methylated mRNAs. *Elife*, **2017**, 6, e31311. PMC5648532.

Reviewer #1 (Remarks to the Author):

The authors have addressed the points I made in my initial review to my satisfaction, and so I support acceptance in its current form.

Reviewer #1 (Remarks on code availability):

I did not attempt to run the code but based on previous experience this should provide sufficient information for the community to assess and reproduce the results as needed.

Reviewer #2 (Remarks to the Author):

The authors have satisfactorily addressed my questions in the revised manuscript.

Reviewer #3 (Remarks to the Author):

All previously raised concerns have been addressed satisfactorily in the revised manuscript and I recommend publication of the study in Nature Communications.